# Diurnal changes in the efficiency of information transmission at a sensory synapse

José Moya-Díaz[1], Ben James[1], Federico Esposti[1], Jamie Johnston [1] & Leon Lagnado [1✉]

Neuromodulators adapt sensory circuits to changes in the external world or the animal's internal state and synapses are key control sites for such plasticity. Less clear is how neuromodulation alters the amount of information transmitted through the circuit. We investigated this question in the context of the diurnal regulation of visual processing in the retina of zebrafish, focusing on ribbon synapses of bipolar cells. We demonstrate that contrast-sensitivity peaks in the afternoon accompanied by a four-fold increase in the average Shannon information transmitted from an active zone. This increase reflects higher synaptic gain, lower spontaneous "noise" and reduced variability of evoked responses. Simultaneously, an increase in the probability of multivesicular events with larger information content increases the efficiency of transmission (bits per vesicle) by factors of 1.5-2.7. This study demonstrates the multiplicity of mechanisms by which a neuromodulator can adjust the synaptic transfer of sensory information.

[1] Sussex Neuroscience, School of Life Sciences, University of Sussex, Brighton BN1 9QG, UK. ✉email: l.lagnado@sussex.ac.uk

It has long been understood that the flow of signals through neural circuits is adjusted by neuromodulators[1]. How does this plasticity translate into changes in the amount of information that is transmitted through the circuit? Here we investigate this question in the retina as visual processing adjusts on a diurnal cycle.

The retina is highly plastic: the input-output relation can adapt within seconds to the recent history of the visual stimulus[2,3] or, on longer time-scales, to changes in the animal's internal state[4,5]. In diurnal animals, for instance, retinal sensitivity to light is regulated both by the daily light-dark cycle and by intrinsic circadian clocks that act on both outer and inner retinal circuitry[6–9]. Key to these adjustments is dopamine, a neuromodulator which is released from amacrine cells in a circadian cycle, varying from a minimum at night, increasing during the day and peaking before dusk[6,10,11]. But the average luminance of a visual scene is not the variable driving most behaviours related to vision: navigation, finding food and avoiding predators all depend on the detection of fast changes in light intensity. We therefore investigated the diurnal control of temporal contrast processing, focusing on the visual signal that bipolar cells transmit to the inner retina.

The synaptic compartments of bipolar cells represent an information bottleneck in vision because they are the only route for transmission of signals originating from photoreceptors. As a result, these compartments are an important control point for transformations of the visual signal[12] and contribute to a number of processing tasks, from adaptive gain control to temporal filtering and the coding of motion, colour, orientation and direction[3,13–15]. In common with other sensory neurons, such as photoreceptors, hair cells and electroreceptors, bipolar cells transmit information through ribbon synapses containing specialized structures that supply vesicles to the active zone[16]. These sensory synapses do not always operate as Poisson machines in which vesicles are released independently but also signal through multivesicular release (MVR), where the fusion of two or more vesicles is co-ordinated as a single synaptic event[17–19]. The importance of MVR at a number of sites in the brain is now recognized and it has been suggested that it might contribute to more complex strategies for transmitting information than modulation of a rate code[20–22].

Shannon's information theory has been used to measure the amount of information carried by neurons using spikes, but these are not the neural events that transmit information across the synapse: there the essential symbol is the quantum of neurotransmitter released from a vesicle. To understand the information leaving a neuron the experimenter therefore needs to observe the fusion of vesicles conveying the message[23,24]. This has recently been achieved by multiphoton imaging of the glutamate reporter iGluSnFR[25] in the retina of larval zebrafish, where it is found that bipolar cells do not transmit the visual message using a simple binary code but instead use a number of symbols formed by one, two, three or more vesicles released as one event[19].

Here we demonstrate that the strategy of synaptic coding by amplitude as well as rate is under diurnal control. The Shannon information transmitted at each active zone increases four-fold in the afternoon compared to the morning and dopamine contributes to this increase by increasing synaptic gain, lowering spontaneous noise and reducing the variability of evoked responses. All three mechanisms operate in the OFF channel signalling decreases in light intensity, but only the last two in the ON channel signalling increases. Crucially, dopamine also adjusts the strategy by which these synapses code visual information by increasing the probability of multivesicular events with larger information content. Larger events carry more bits of information per vesicle so dopamine also increases the efficiency of the vesicle code.

## Results

**Differential regulation of luminance sensitivity and contrast sensitivity**. To investigate the diurnal modulation of visual processing in the retina of zebrafish we began by imaging activity of the terminals of bipolar cells with a synaptically localized calcium reporter, SyGCaMP2[26] (Fig. 1A). When animals were placed on a cycle of 14 h light and 10 h dark, no significant synaptic responses could be detected in the 6 h preceding light onset (*Zeitgeber* times 18–0 h), consistent with previous observations that larvae are blind at night[27]. Visual sensitivity began to recover within 20 min of light onset, after which responses gradually increased in amplitude (Fig. S1A and Fig. 1B). Plotting the luminance-response functions (Fig. 1C) allowed the light sensitivity to be quantified as the inverse of the luminance generating a half-maximal response ($1/I_{1/2}$). Over the course of the day, luminance sensitivity increased gradually over a range greater than 200-fold (Fig. 1D). As in other species, this increase could be explained largely by the actions of D2 dopamine receptors because injection of the antagonist sulpiride (~2 μM) reduced luminance sensitivity in the afternoon to levels measured in the morning[6] (Fig. S1).

The detection of modulations in light intensity was also under diurnal control, but with a different time course (Fig. 1E–G; 5 Hz full-field stimuli). At ZT = 4 h, temporal contrasts below 50% were barely detected and the half-maximal response ($C_{1/2}$) was generated by a contrast of 86 ± 2 % (Fig. 1E, F). But at ZT = 7 hours $C_{1/2}$ fell to 35 ± 2 % with responses saturated above 50% contrast. When contrast sensitivity ($1/C_{1/2}$) was mapped during the course of the day it was relatively constant at ZT 1–5 h and ZT 9–14 h but increased to levels ~2.4-fold higher around ZT = 7 h (Fig. 1G). Notably, this peak in the contrast sensitivity of the retinal circuit occurred at a similar *Zeitgeber* time as the maximum contrast sensitivity measured behaviourally using the optokinetic reflex[9,28]. A qualitatively similar increase in contrast sensitivity was also observed at the retinal output projecting to the optic tectum (Fig. S2).

**Diurnal regulation of contrast gain**. To explore the diurnal regulation of contrast sensitivity we measured transmission of the visual signal to the inner retina in terms of its elementary units— synaptic vesicles—by expressing the glutamate reporter iGluSnFR[25] sparsely in bipolar cells (Fig. 2A). A variety of morphological types of bipolar cell were investigated but we focused on a comparison of the two most basic functional groups, ON and OFF cells, identified through their responses to steps of light (Fig. S3). Wiener deconvolution of iGluSnFR signals allowed us to count released vesicles and evidence that these methods allow the isolation of glutamatergic signals from individual active zones has been described[19] (Fig. S4). Synaptic function was compared between a 2-h period centred on ZT = 1 h ("morning") and a 2-h period centred on ZT = 7 h ("afternoon"; Fig. 1G).

Examples of glutamate transients at the active zone of an OFF bipolar cell are shown in Fig. 2B. Across a range of contrasts, responses were, on average, larger in the afternoon. The contrast-response function (CRF) was constructed simply by measuring the average number of vesicles released per cycle of a 5 Hz stimulus (full-field), choosing this frequency because the integration time of a bipolar cell is ~200 ms[29]. There was little diurnal modulation of the CRF measured at ON synapses but in the OFF channel the average synaptic gain, measured as the maximum rate of vesicle release at 100% contrast, increased from 15.25 ± 2.5 vesicles/s in the morning to 25.5 ± 1.5 vesicles/s in the afternoon (Fig. 2C, D).

Increases in synaptic gain were not simply multiplicative but also accompanied by an increase in contrast sensitivity. To assess these combined effects we calculated the derivative of the CRF,

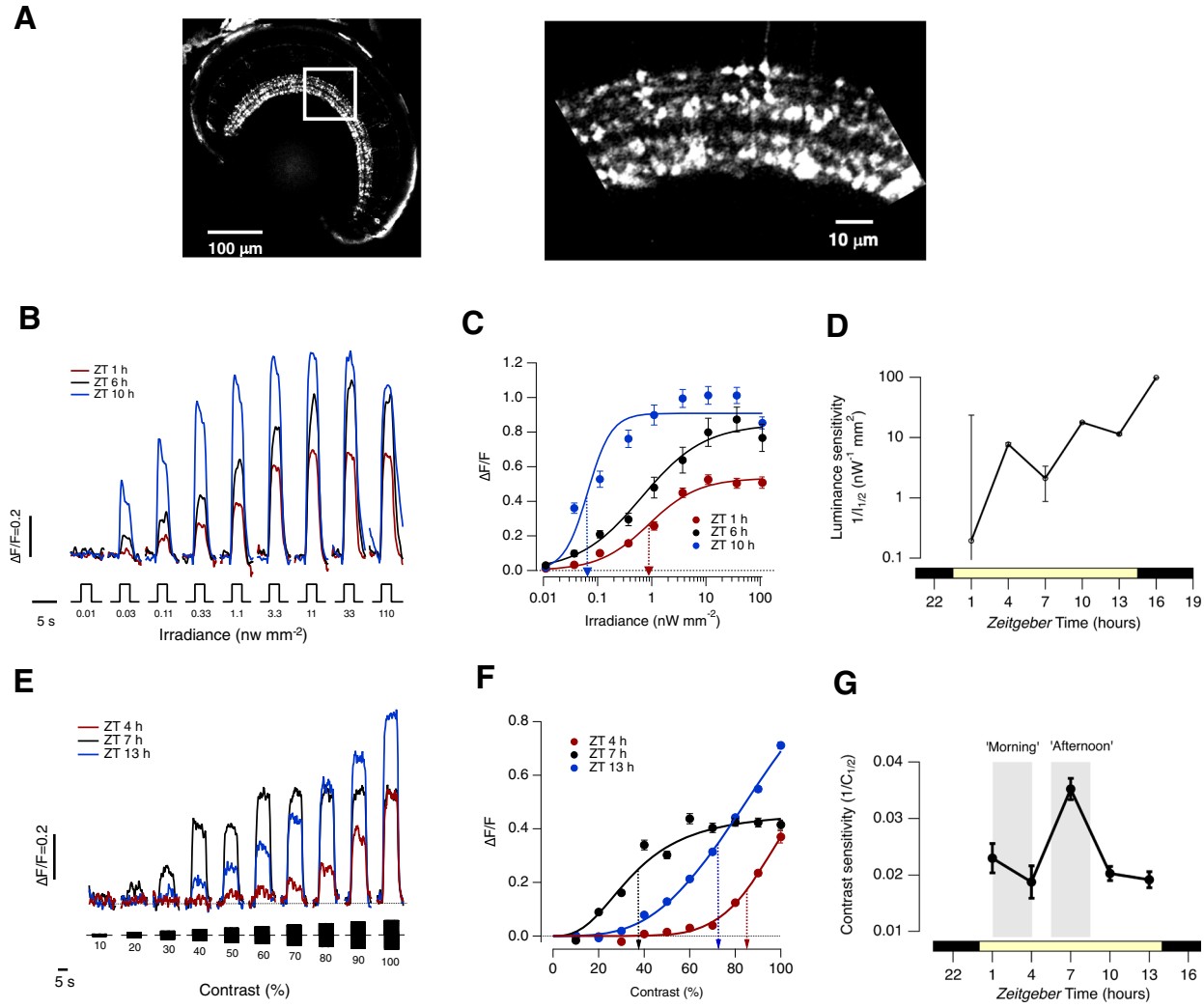

**Fig. 1 Differential regulation of luminance sensitivity and contrast sensitivity. A** Left: Retina of a Ribeye:SyGCaMP2 fish with box over the inner plexiform layer (IPL). Right: expansion of the boxed region showing terminals of bipolar cells. Zebrafish larvae were 7–9 days post-fertilization. **B** Averaged responses from ON terminals to light steps of different irradiance measured at *Zeitgeber* time 1, 6 and 10 hours. Note large variations in amplitude and kinetics. The full-field light stimuli were generated by an amber LED ($I_{max} = 590$ nm) which will most effectively stimulate red and green cones. Each light step lasted 3 s ($n = 535$ terminals from 10 fish). **C** Peak response as a function of irradiance for ON terminals in (**B**). The smooth lines are Hill functions of the form $R = R_{max}*(I^h/(I^h + I_{1/2}^h))$, where $R$ is the peak response, $I$ is the irradiance, $h$ is the Hill coefficient and $I_{1/2}$ is the irradiance generating the half-maximal response. At ZT = 6 h: $R_{max} = 0.91 \pm 0.04$; $h = 2.0 \pm 0.2$; $I_{1/2} = 0.066 \pm 0.02$ nW/mm$^2$ (dashed blue arrow). At ZT = 10 h: $R_{max} = 0.85 \pm 0.06$; $h = 0.8 \pm 0.1$; $I_{1/2} = 0.65 \pm 0.18$ nW/mm$^2$. At ZT = 1h: $R_{max} = 0.853 \pm 0.02$; $h = 0.9 \pm 0.2$; $I_{1/2} = 0.88 \pm 0.18$ nW/mm$^2$ (dashed red arrow). **D** Variations in luminance sensitivity as a function of *Zeitgeber* time averaged across both ON and OFF terminals ($n = 535$ and 335 terminals, respectively). The lower bar shows the timing of the light-dark cycle. Error bars are ± 1 SD. **E** Averaged responses to stimuli of different contrasts (i.e. sinusoidal modulations in light intensity around a mean) measured at *Zeitgeber* time 4, 7 and 13 h averaged across both ON and OFF terminals ($n = 949$ from 21 fish). **F** Peak response amplitude as a function of contrast for terminals shown in E. The smooth lines are Hill functions used to interpolate values of $C_{1/2}$, the contrast generating the half-maximal response. Note the diurnal variations. At ZT = 4 h: $C_{1/2} = 86 \pm 2\%$ (dashed red arrow); $h = 7.0 \pm 1.2$. At ZT = 7 h: $C_{1/2} = 35 \pm 2\%$ (dashed black arrow); $h = 2.7 \pm 0.2$. At ZT = 13 h: $C_{1/2} = 72 \pm 2\%$; $h = 3.3 \pm 0.2$ (dashed blue arrow). **G** Variations in contrast sensitivity as a function of *Zeitgeber* time averaged across ON and OFF terminals ($n = 949$ from 21 fish). Note the peak around ZT = 7 h which is not mirrored in the diurnal variation in luminance sensitivity (**D**). The grey bars show the periods described as "morning" and "afternoon". All error bars show ± 1 s.e.m. except for (**D**) which is ± 1 SD. Source data are provided as a Source Data file.

which we term "contrast gain" (Fig. 2E). A second reason for re-expressing the contrast-response functions as "contrast gain" is that this gives insight into a key property of the visual system—its ability to discriminate one stimulus from another[30,31]. Contrasts in natural visual scenes rarely exceed 40%[14] and in the morning changes in this range were signalled best through the ON channel (grey box in Fig. 2E). But in the afternoon the OFF channel became dominant, with contrast gains increasing by factors of 2–6 at contrasts up to 40%. Diurnal modulation of retinal

processing therefore altered the relative importance of ON and OFF pathways in signalling temporal contrast.

**Dopamine regulates contrast gain.** Dopamine is a key regulator of the luminance sensitivity of the retina[6,10,11]. To test whether this neuromodulator also adjusts contrast sensitivity we injected agonists or antagonists of dopamine receptors directly into the eye. Figure 3A shows examples of the output from a synapse imaged in the afternoon, before and after injection of the D1

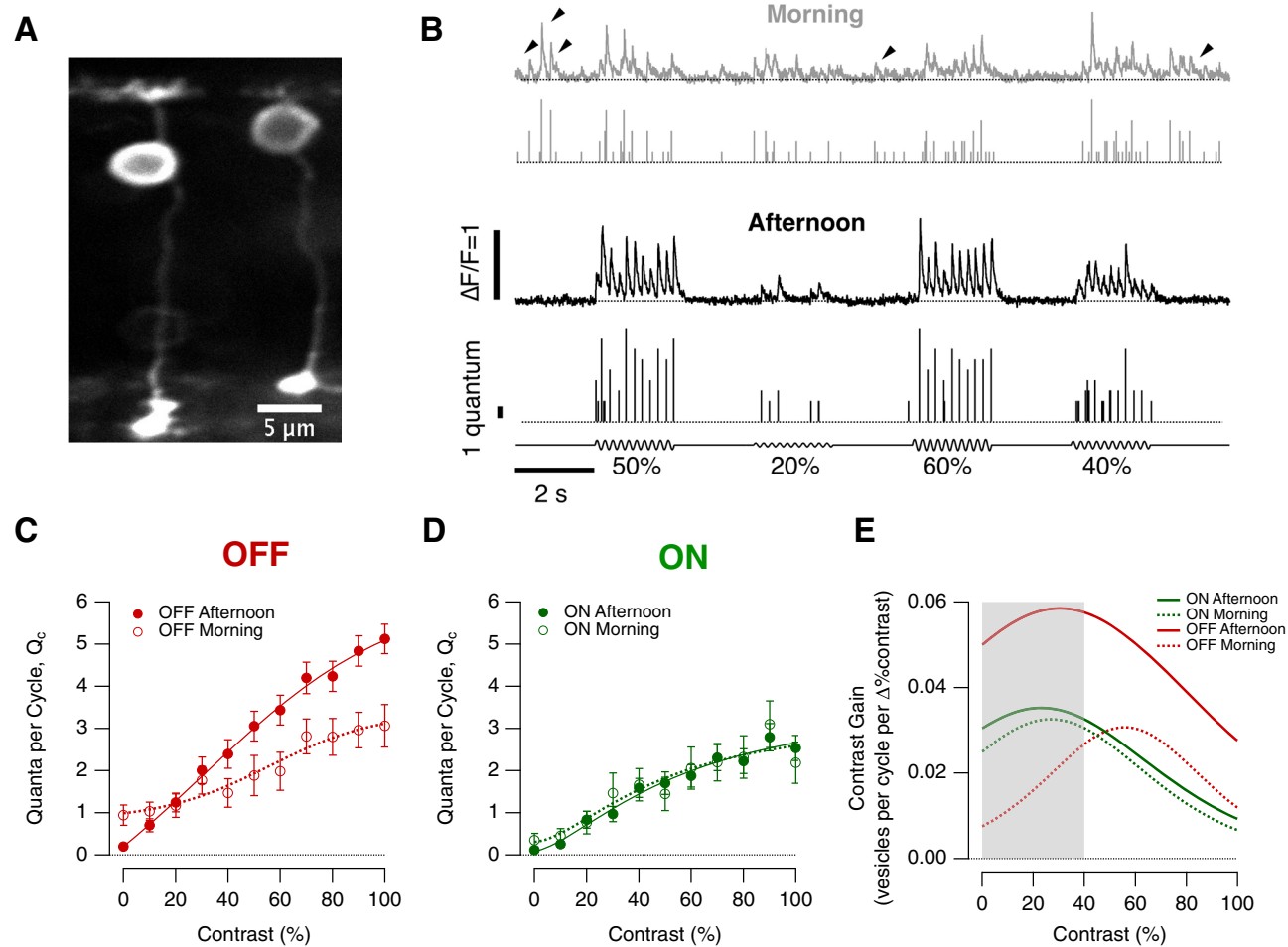

**Fig. 2 Diurnal modulation of synaptic gain. A** Multiphoton section through the eye of a zebrafish larva (7 dpf) expressing iGluSnFR in a subset of bipolar cells. **B** Examples of iGluSnFR signals from an individual OFF synapse elicited using stimuli of variable contrast modulated at 5 Hz (0–100%, full field, sine wave) in the morning (ZT 0–2 h, grey) and afternoon (ZT 6–8 h, black). Note the high levels of spontaneous activity in the morning (black arrowheads). In each case the top trace shows the iGluSnFR signal and the lower trace the estimated number of quanta composing each event ($Q_e$). **C** Average contrast-response functions in OFF bipolar cell synapses in the morning (open circles; $n = 20$ synapses) and afternoon (closed; $n = 59$), where the response ($R$) was quantified as the average of quanta per cycle, $Q_c$ (i.e., the total number of quanta released *within* a single cycle of the sinusoidal stimulus). Each point shows the mean ± s.e.m. The smooth lines are fits of a sigmoid used for smoothing. Note the differences in the shape of the contrast-response functions and in the levels of spontaneous activity (zero contrast) (One-way ANCOVA test, $p < 0.0006$). **D** Average contrast-response functions in ON bipolar cell synapses in the morning (open circles; $n = 12$ synapses) and afternoon (closed; $n = 31$). There was no significant difference in the morning relative to afternoon. (One-way ANCOVA test, $p = 0.53$) Each point shows the mean ± s.e.m. **E** The contrast gain calculated as the derivative of the fits to the contrast-response functions in (**C**, **D**). The grey box provides an indication of the contrasts most common in nature (below about 40%). Note that the maximum contrast discrimination is increased by a factor of 2x in the OFF channel during the afternoon. Source data are provided as a Source Data file.

receptor antagonist SCH 23390 (estimated final concentration of 20 nM). Counteracting the actions of endogenous dopamine reduced the average rate of vesicle release and shifted the CRF such that the maximum contrast gain was achieved at higher levels (black points in Fig. 3B, C). Conversely, increasing activation of D1 receptors in the morning by injection of the agonist ADTN (~0.2 μM) increased response gain.

The dynamic range over which D1 receptors adjusted synaptic gain was calculated as the ratio of the CRFs in the presence of the agonist and antagonist ("relative response gain"): in both ON and OFF channels the maximum modulation was ~16-fold, occurring at contrasts of 20–40% (Fig. 3D). Diurnal modulation of gain was narrower than this potential range, with a maximum of 1.7-fold in OFF synapses. This difference reflected, at least in part, a gain in the morning that was at least 5-fold higher than that measured with D1R receptors blocked, consistent with dopamine levels that were already high enough to potentiate synaptic transmission

(Fig. 3B, C). These manipulations of retinal dopamine receptors caused qualitatively similar changes in the signals that ganglion cells transmit to the optic tectum (Fig. S2).

**Modulation of synaptic noise and variability**. We next asked how diurnal changes in the operation of the retina affected the amount of information that bipolar cells transmit to post-synaptic ganglion cells and amacrine cells. In the framework of information theory[23], an increase in synaptic gain will tend to increase information about a change in contrast by causing a larger change in the average number of vesicles released. But the gain with which a synapse responds to a sensory signal is not the only property that determines information transmission: neural information is degraded by "noise" that causes responses to vary when the same stimulus is repeated and synapses are a major source of such variability[32,33]. Synaptic noise is an inevitable consequence of the stochasticity of the presynaptic processes that

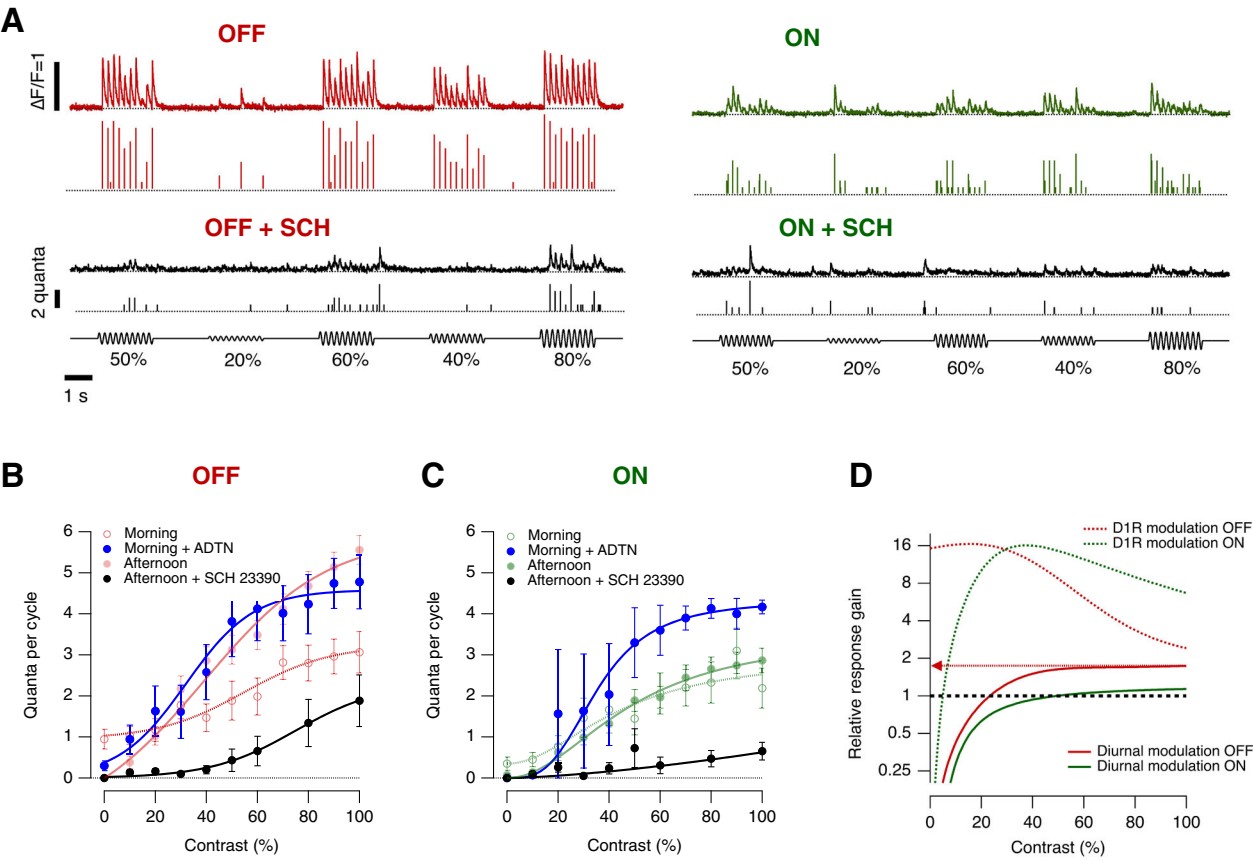

**Fig. 3 Diurnal changes in dopamine levels modulate synaptic transmission. A** Examples of iGluSnFR signals recorded in the afternoon from an individual OFF (red trace) and ON (green trace) synapses elicited using a stimulus of variable contrast before and after intravitreal injection of the D1 antagonist, SCH 23390 (black traces; estimated final concentration 20 nM). Note that SCH 23390 abolished synaptic responses at lower contrasts in ON and OFF synapses. In each case the top trace shows the iGluSnFR signal and the lower trace the estimated $Q_e$. **B** Average contrast-response functions in OFF bipolar cell synapses after administration of D1 antagonist (black dots) in the afternoon and after administration of the D1 agonist ADTN in the morning (blue dots). Each point shows the mean ± s.e.m. (SCH 23390, $n = 12$ synapses; ADTN, $n = 12$ synapses). Control responses observed in the morning and afternoon are superimposed (red dots, from Fig. 2C). **C** Average contrast-response functions in ON bipolar cell synapses after intravitreal injection of D1 antagonist in the afternoon (black dots) and ADTN in the morning (blue dots). Each point shows the mean ± s.e.m. (SCH 23390, $n = 7$ synapses; ADTN, $n = 5$ synapses). Control responses observed in the morning and afternoon are superimposed (green dots, from Fig. 2D). **D** Relative response gain by diurnal modulation and after manipulation of dopaminergic signalling (dashed lines). Note that diurnal modulation of synaptic gain is higher in OFF synapses, whereas dopamine modulates the dynamic range by ~16-fold-change in ON and OFF synapses. Source data are provided as a Source Data file.

control the fusion of vesicles[20] and such variability was a prominent feature of the output from synapses of bipolar cells, as illustrated in Figs. 2B and 3A.

We distinguished four aspects of synaptic noise and investigated the diurnal modulation of each; (i) spontaneous vesicle release (Fig. 4), (ii) variability in the number of vesicles released by a repeated stimulus (Fig. 5), (iii) variability in the timing of release events i.e how tightly they are time-locked to the stimulus (Fig. 6) and (iv) changes in the distribution of event amplitudes i.e modulation of multivesicular release (Fig. 7).

Finally, we calculated how these different aspects of synaptic noise combined with changes in contrast gain (Fig. 3) to alter the amount of visual information transferred from individual active zones (Fig. 8).

*Spontaneous release.* Increases in synaptic gain were accompanied by a *decrease* in the spontaneous release of vesicles in the absence of a visual stimulus. In the morning, spontaneous events occurred at relatively high rates, some composed of single vesicles and others of two or more (Fig. 4A–C). Integrating across events of all amplitudes, the average rate of spontaneous release in OFF synapses was $1.95 \pm 0.04$ vesicles s$^{-1}$ in the morning, falling to

$0.37 \pm 0.02$ vesicles s$^{-1}$ in the afternoon (Fig. 4B; significant at $p < 0.0001$, one-way ANOVA). In ON synapses these values were $1.47 \pm 0.09$ vesicles s$^{-1}$ and $0.46 \pm 0.02$ vesicles s$^{-1}$ ($p < 0.0001$, Fig. 4C). Across both channels, therefore, spontaneous noise was 3–5 times lower in the afternoon compared to the morning.

Spontaneous noise in OFF synapses was modulated by dopamine. Increased activation of D1 receptors by injection of ADTN suppressed spontaneous release in the morning to levels close to those measured in the afternoon ($0.51 \pm 0.03$ vesicles s$^{-1}$), a change significant at $p < 0.0001$ (One-way ANOVA; Fig. 4B). In contrast, ADTN had no significant effect on spontaneous release from ON synapses ($1.66 \pm 0.18$ vesicles s$^{-1}$, $p = 0.3$; Fig. 4C).

*Variability in stimulus-evoked responses.* The reliability of neural responses measured as spikes can be expressed using the Fano factor: the ratio of the variance-to-mean of spikes counted in a fixed time-window after a repeated stimulus[34,35]. We used a similar approach to assess the reliability of synapses, calculating the Fano factor ($F$) by counting the number of vesicles released over each cycle of a sinusoidal stimulus (Fig. 5A). Through both ON and OFF channels, the variability of synaptic output was significantly higher than expected for a Poisson process, for which

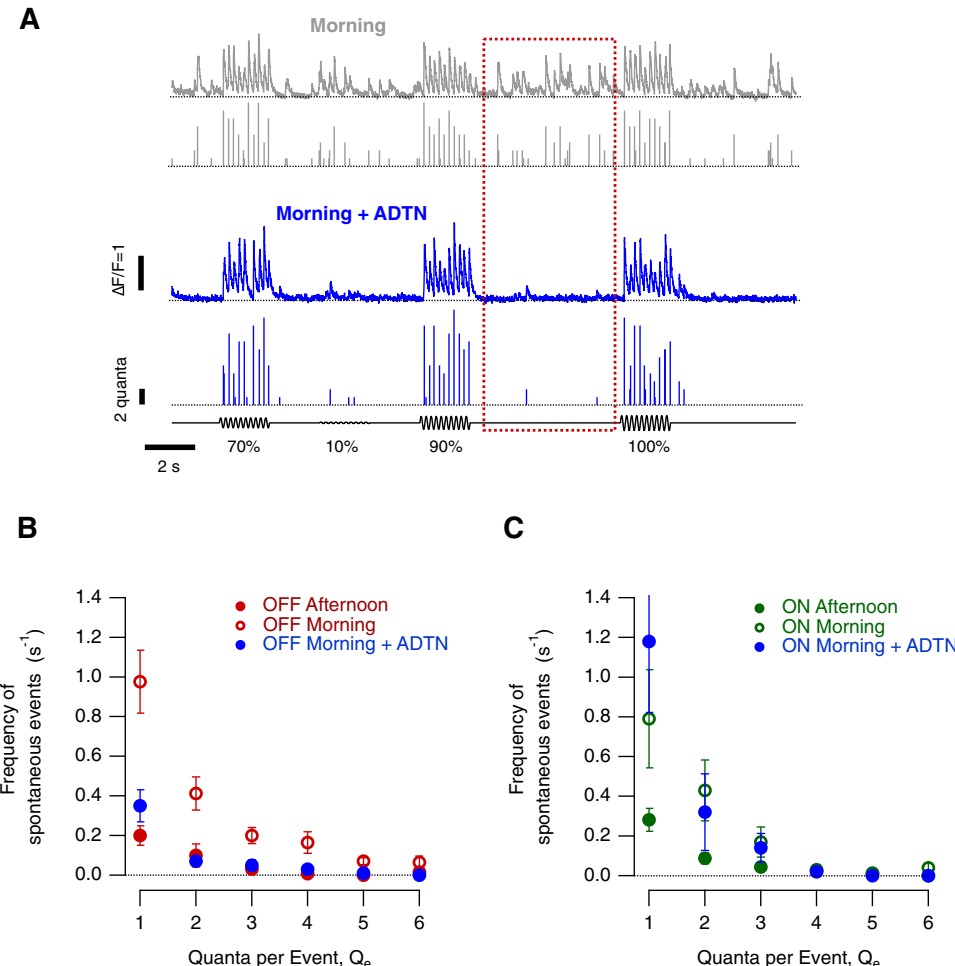

**Fig. 4 Diurnal modulation of spontaneous synaptic noise. A** *Top:* Example of iGluSnFR signals from an individual OFF synapse elicited using a stimulus of variable contrast in the morning (0–100%, 5 Hz modulation). In this example, note the high levels of spontaneous activity that were quantified as the responses elicited at zero contrast (red dashed box). *Bottom.* Examples of iGluSnFR signals from the same OFF synapse after intravitreal injection of ADTN. Note the increase in amplitude and frequency of events and the reduction of spontaneous activity. In each case the top trace shows the iGluSnFR signal and the lower trace the estimated $Q_e$. **B** Quantification of spontaneous events composed by different $Q_e$ in OFF synapses in the morning, morning + ADTN and afternoon (OFF morning, $n = 20$ synapses; OFF morning + ADTN $n = 12$ synapses; OFF afternoon, $n = 24$ synapses). Note the suppression of spontaneous events in OFF synapses after intravitreal injection of ADTN in the morning. Each point shows the mean ± s.e.m. **C** Quantification of spontaneous events composed by different $Q_e$ in ON synapses in the Morning, Morning + ADTN and Afternoon (ON Morning $n = 12$ synapses; ON Morning + ADTN $n = 5$ synapses; ON Afternoon, $n = 17$ synapses). Note that spontaneous activity levels were not dramatically altered after administration of ADTN. Each point shows the mean ± s.e.m. Source data are provided as a Source Data file.

the Fano factor is one (Fig. 5B, C). This yields a fundamental insight: the ribbon synapses of bipolar cells do not adhere to the common model of synaptic function in which all vesicles are released independently.

In the morning, F was ~2.7 in both ON and OFF synapses when averaged over contrasts of 10-100%, but synapses were more reliable in the afternoon, with F falling to ~1.6 (One-Way ANOVA, $p < 0.0001$; Fig. 5B–E). In the OFF channel, the increase in contrast gain and sensitivity in the afternoon was therefore also associated with increased reliability of bipolar cell synapses. Although synapses in the ON channel did not undergo significant changes in gain and sensitivity, they also became more reliable. These diurnal changes in synaptic reliability were also dopamine-dependent, activation of D1 receptors in the morning reducing F to levels similar to those measured in the afternoon (Fig. 5D, E).

*Temporal jitter.* Retinal ganglion cells (RGCs) encode information not just in their spike count but also in the timing of spikes[34,36]. Spike times can vary by just a few milliseconds and this accuracy

depends on the precision of excitatory inputs received from bipolar cells[37]. The standard deviation in timing of release events ("temporal jitter") was measured relative to the phase of a 5 Hz stimulus (60% contrast; Fig. 6A) and the larger the release events the more precise it was on average (Fig. 6B, C). In OFF synapses the temporal jitter was 5–8 ms higher in the morning compared to the afternoon for events composed of up to 8 vesicles (Fig. 6B; $p < 0.008$, Kolomogorov–Smirnov test). Diurnal modulation of temporal precision was weaker in ON synapses and only significant for events composed of 1–3 vesicles (Fig. 6C; $t$ test at each $Q_e$). Increasing activation of D1 receptors in the morning reduced temporal jitter in events composed of multiple quanta in OFF synapses ($p < 0.05$; KS test) but not ON (Fig. 6B, C; $p > 0.5$). Diurnal variations in dopamine therefore modulate the temporal accuracy of vesicle release in OFF synapses.

*Changes in the distribution of multivesicular events.* Previous studies quantifying the synaptic transfer of visual information have been limited by the inability to monitor individual active

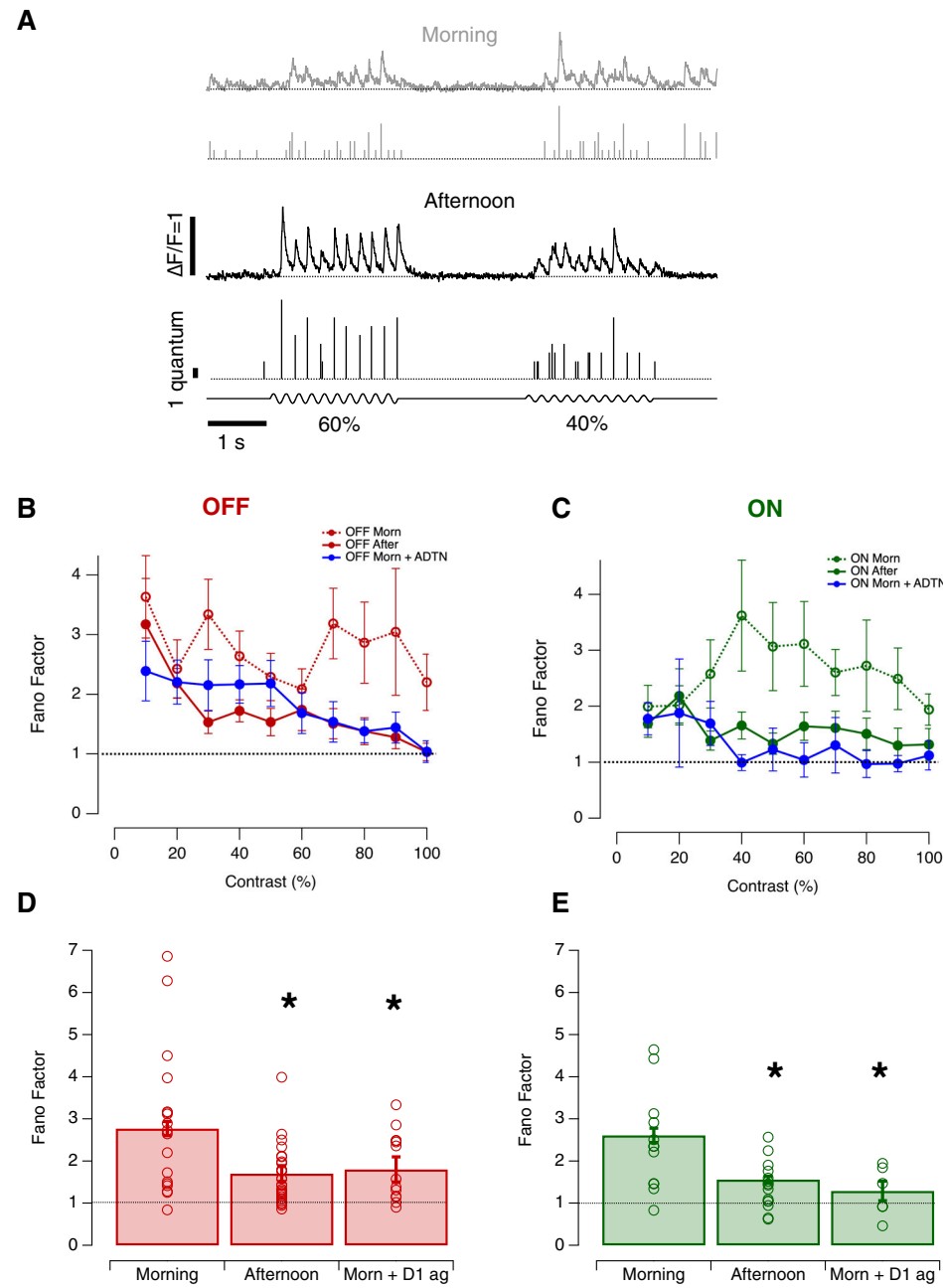

**Fig. 5 Diurnal changes in the variability of stimulus-evoked responses. A** Examples of iGluSnFR signals from individual OFF synapses in the morning and afternoon. Responses elicited by stimuli of 60% and 40% contrast varied from cycle to cycle of the 5 Hz stimulus. In each case the top trace shows the iGluSnFR signal and the lower trace the estimated $Q_e$. **B** Variability in the response of OFF synapses calculated as the Fano factor, with each response measured as the total number of vesicles released over one cycle at each contrasts. Comparison is made between the morning ($n = 18$), afternoon ($n = 27$) and the morning after injection of ADTN ($n = 13$). Each point shows the mean ± s.e.m. **C** As in B, but for ON synapses ($n = 12$, 15 and 6 synapses for respective conditions). **D** Average Fano factor over different contrasts in OFF synapses in the three conditions described above. Data from (**B**, **C**). Overall, the average Fano factor was significantly higher in the morning compared to afternoon or in the morning after injection of ADTN (One-Way ANOVA; $p < 0.0001$). Open red dots represent individual values. Error bars show ± s.e.m. **E** As (**D**), but for ON synapses. Again, the average Fano factor was significantly higher in the morning (One-Way ANOVA; $p < 0.0001$). Open green dots represent individual values. Error bars show ± s.e.m. Source data are provided as a Source Data file.

zones and used the assumption that vesicles are released according to Poisson statistics[38,39]. But we now know that bipolar cells do not employ a simple rate code and visual information is also contained in the *amplitude* of multivesicular events[19]. We therefore tested whether modulation of contrast gain was accompanied by changes in the way stimulus contrast was encoded by the distribution of $Q_e$, the number of quanta in an event, and found that it was.

A comparison of the distribution of $Q_e$ in the morning and afternoon is shown in Fig. 7A for responses to a stimulus of 60% contrast. In ON synapses, 68% of release events in the morning were univesicular, falling to 40% in the afternoon as the distribution shifted towards larger MVR events (Fig. 7B; $p < 0.05$, KS test). This shift was fully reversed by antagonizing D1 receptors by injection of SCH 23390 (Fig. 7C; $p < 0.001$). In the morning, MVR was more prevalent in OFF synapses than

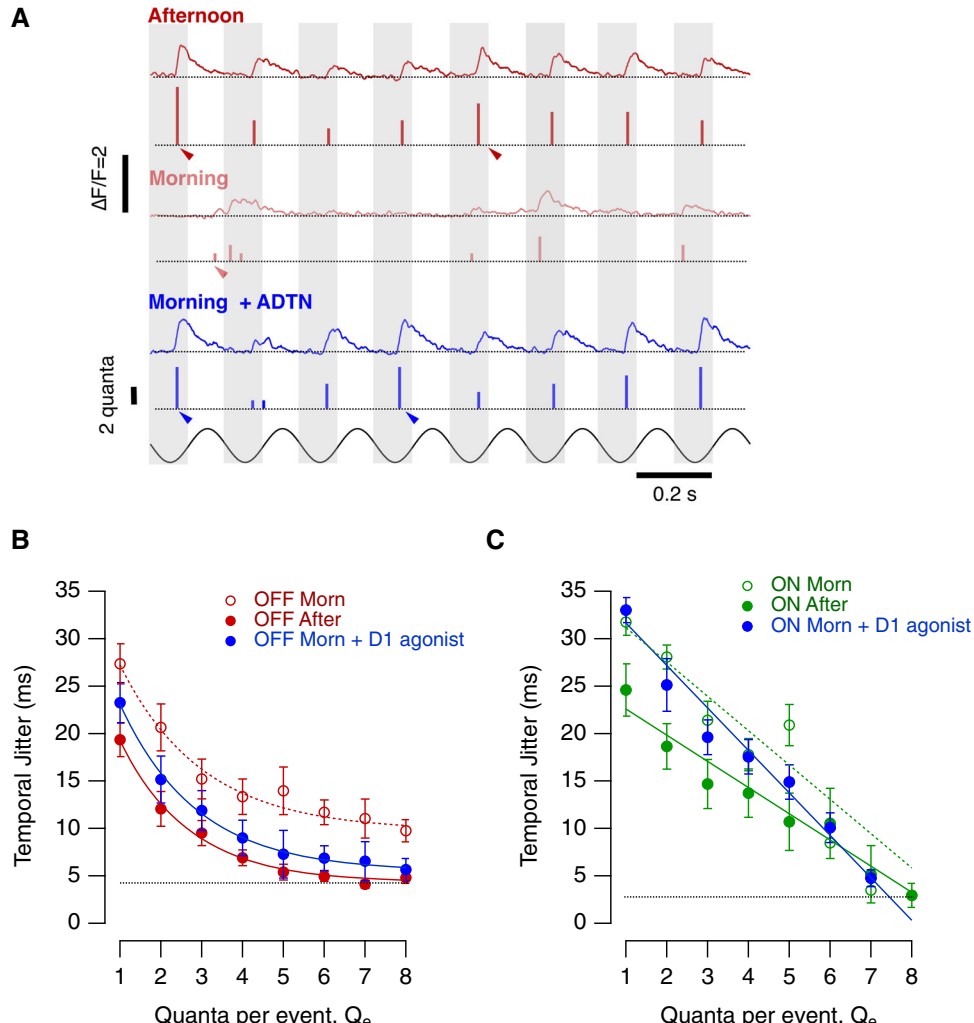

**Fig. 6 The temporal precision of vesicle release is under diurnal control in the OFF channel. A** Example recordings from two OFF synapses stimulated at 60% contrast in three conditions: afternoon (top, red trace), morning (middle, light red trace) and after intravitreal injection of ADTN in the morning (bottom, blue trace). Morning and morning + ADTN synaptic responses are from the same synapse. The modulation in intensity (5 Hz, sine wave) is shown below. Arrowheads highlight events occurring at different phases of the stimulus, with less variation with events composed for 4 or more quanta in the afternoon and after administration of ADTN in the morning. In each case the top trace shows the iGluSnFR signal and the lower trace the estimated $Q_e$. **B** Temporal jitter of events composed of different numbers of quanta in OFF synapses in the afternoon (red dots; $n = 24$ synapses); Morning (open red dots; $n = 19$ synapses) and Morning + ADTN (blue dots, $n = 16$). Note that during the morning events composed by multiple quanta were less phase-locked to the stimuli in comparison to the afternoon. Activation of D1 receptors had a significant effect on release of multiquantal events. Events composed by 5 or more quanta jittered by ~5 ms, similar to values observed in the afternoon (black dashed line). The solid lines describing these relations in the three conditions are better described by a single exponential decay function of the form $y_0 + A_{\exp}((-(x-x_0)/\tau)))$ with $y_0 = 4.23 \pm 1.2$ and $A = 27 \pm 7$ in the afternoon; $y_0 = 9.77 \pm 1.4$ and $A = 28.64 \pm 5.6$ in the morning and $y_0 = 5.45 \pm 1.3$, $A = 30 \pm 6.1$ after activation of D1 receptor in the morning. Each point shows the mean ± s.e.m. **C** Temporal jitter of events composed by different numbers of quanta measured in ON synapses in the afternoon (green dots; $n = 14$ synapses) during the morning (open green dots; $n = 10$ synapses) and during Morning + ADTN, (blue dots; $n = 6$ synapses). Activation of D1 receptor did not have a significant effect in the temporal precision in the ON channel. The relationships observed in the different conditions were better described by a straight line. Morning $a = 34.7 \pm 1.5$ and a slope $= -3.6 \pm 0.5$; Afternoon: $a = 25.1 \pm 1.2$ and a slope $= -2.8 \pm 0.2$; Morning + ADTN: $a = 34.9 \pm 1.5$ and a slope $= -4.2 \pm 0.3$. Note that events composed of 8 or more quanta jittered by just ~4 ms (black dashed line). Each point shows the mean ± s.e.m. Source data are provided as a Source Data file.

ON, with only 38% of release events being univesicular but again there was a significant shift towards larger events in the afternoon (Fig. 7D; $p < 0.02$). Blocking the D1 actions of endogenous dopamine had a stronger effect in OFF synapses, increasing the proportion of univesicular events to 66% in the afternoon (Fig. 7E; $p < 0.001$). Qualitatively similar modulation of MVR was observed over a range of contrasts from 20% to 80% and blocking D1 receptors in the afternoon shifted the distribution back to univesicular release in both ON and OFF channels.

**Modulation of information encoded at the synapse.** How do changes in synaptic gain (Figs. 2 and 3), variability (Figs. 4–6) and MVR (Fig. 7) combine to alter the amount of visual information transmitted by the synapses of bipolar cells? A larger signal relative to noise will tend to increase the mutual information (I) between the response (Q) and the stimulus generating it (S), although the size of the increase will depend on the statistical properties of both signal and noise[40]. But how should we quantify the synaptic signal? When analyzing the spike code, all events

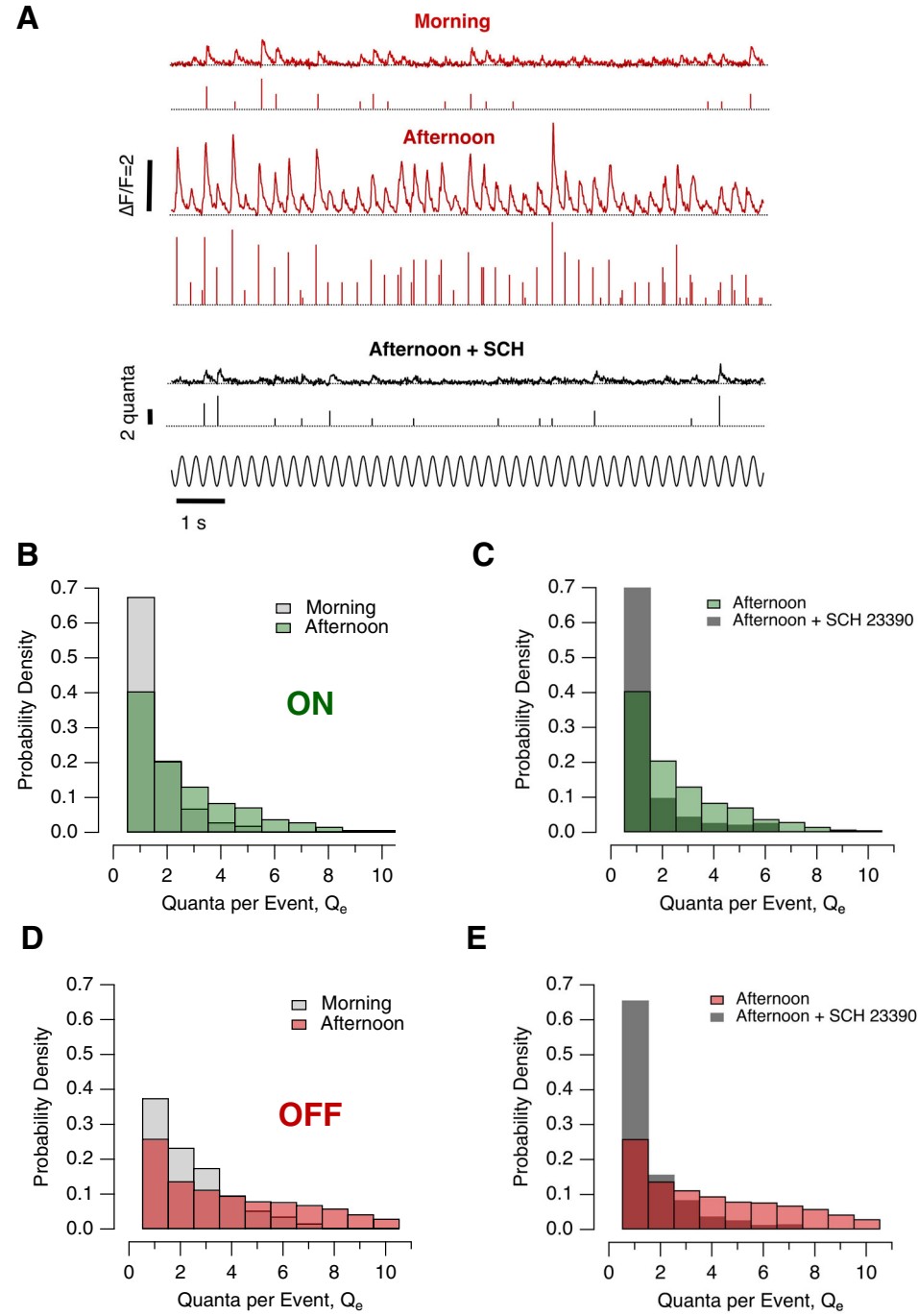

**Fig. 7 Dopamine contributes to diurnal variations in the distribution of multivesicular events. A** Examples of iGluSnFR signals from individual OFF synapses elicited using 60% contrast stimulus (5 Hz, 30 s) in the morning (top), afternoon (middle) and afternoon + SCH 23390 (bottom). In each case the top trace shows the iGluSnFR signal and the lower trace the estimated $Q_e$. **B** Changes in $Q_e$ in ON synapses in the morning (light grey bars, $n = 10$ synapses) and afternoon (green bars, $n = 14$ synapses). In the afternoon the distribution was shifted toward multiquantal events ($p < 0.05$, KS-test). **C** Changes in the distribution of $Q_e$ in ON synapses before and after intravitreal injection of the D1 antagonist SCH23390 (dark grey bars, $n = 8$ synapses). The distribution was shifted toward lower $Q_e$ ($p < 0.001$) but was not significantly different to that measured in the morning. **D** Changes in $Q_e$, in OFF synapses in the morning (light grey bars, $n = 19$ synapses) and afternoon (red bars, $n = 24$ synapses). In the afternoon the distribution was shifted toward multiquantal events ($p < 0.02$). **E** Changes in the distribution of $Q_e$ in OFF synapses before and after intravitreal injection of SCH 23390 in the afternoon (dark grey bars, $n = 12$ synapses). The distribution was shifted toward uniquantal events ($p < 0.001$). Source data are provided as a Source Data file.

comprise the same symbol and the response can be described as the number of spikes in each of a series of time bins[24,40]. The output from bipolar cells is qualitatively different with a visual stimulus being encoded both by the timing of release events and their amplitudes[19]. We therefore took an approach in which MVR events composed of different numbers of vesicles were considered different symbols for the conveying of information[23,41]. The mutual information between the response and stimulus was then computed as the average amount of information about the stimulus gained from observing any symbol (see Methods).

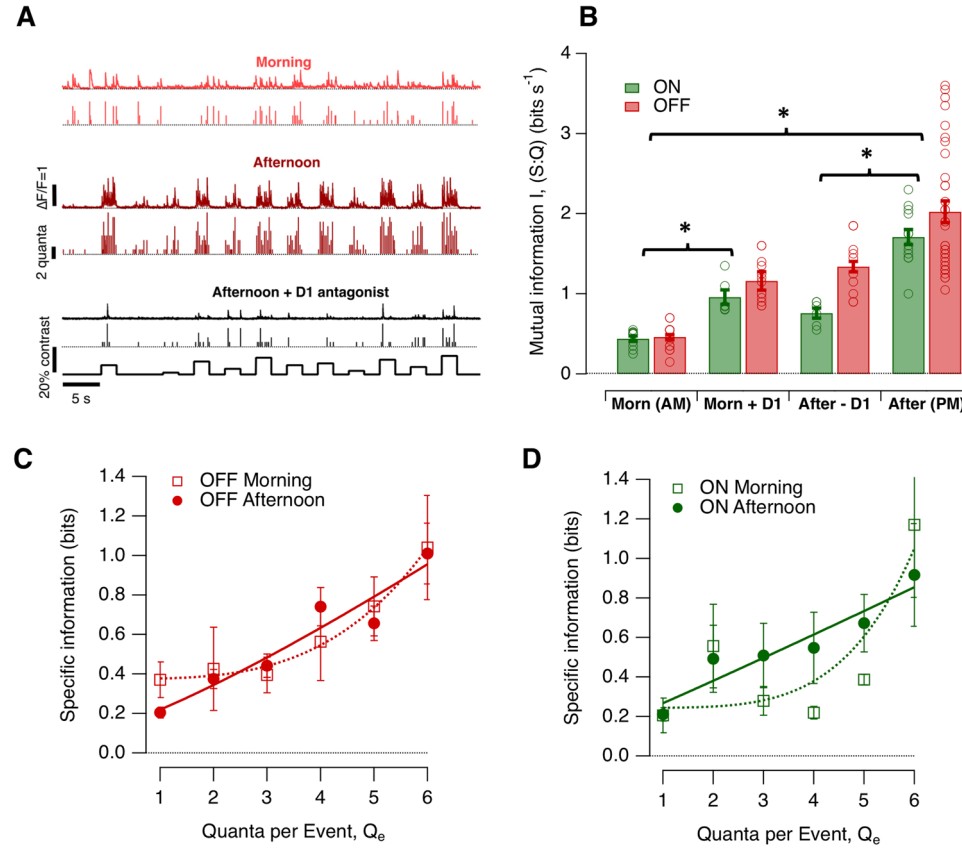

**Fig. 8 Diurnal changes in the efficiency with which synapses transmit visual information. A** Examples of synaptic responses over 11 different contrasts spanning ±10% around the contrast eliciting the half-maximal response ($C_{1/2}$) in the morning (top, light red), afternoon (middle, dark red) and after injection of D1 antagonist SCH 23390 in the afternoon (bottom, black; note the lower frequency and amplitude of release events). In each case the top trace shows the iGluSnFR signal and the lower trace the estimated $Q_e$. Each contrast step lasted 2 s (5 Hz) and each trace is from a different OFF synapse. **B** Mutual information $I (S:Q)$ in four conditions: morning (Morn AM; OFF = 15 synapses, ON = 10 synapses), morning after injection of ADTN (Morn + D1; OFF = 14 synapses, ON = 6 synapses), afternoon after injection of SCH 23390 (After−D1; OFF = 12 synapses, ON = 6 synapses) and afternoon (After PM; OFF = 33 synapses, ON = 13 synapses). Differences between morning and afternoon were significant at $p < 0.0001$ (One-way ANOVA), as were the effects of drug manipulations. Bar graphs show the mean ± s.e.m. Individual values are represented by green and red open dots for ON and OFF synapses, respectively. **C** Specific information ($I$) for events of different quantal content in OFF synapses (morning, $n = 15$; afternoon, n = 33). The curve describing the relation are least-squares fit of a power function of the form $I = y0 + AQ_e^x$. In the morning, $y_0 = 0.38$, $A = 0.0017$, $x = 3.4$. In the afternoon, $y_0 = 0.12$, $A = 0.10$, $x = 1.2$. Each point shows the mean ± s.e.m. **D** As (**C**), but for ON synapses (morning, $n = 10$; afternoon, $n = 13$). In the morning, $y_0 = 0.24$, $A = 0.0003$, $x = 4.4$. In the afternoon, $y_0 = 0.16$, $A = 0.11$, $x = 1.0$. Each point shows the mean ± s.e.m. Source data are provided as a Source Data file.

The stimulus set $S$ comprised 11 different contrasts but these were not fixed for each synapse because the contrast sensitivity varied between synapses and between morning and afternoon (Fig. 1E–G). To make allowance for this, we used contrasts spanning ±10% around $C_{1/2}$ measured within the synapse under study immediately before delivering the stimulus set. In the absence of information about the distribution of contrasts normally experienced by a larval zebrafish, a uniform distribution of contrasts was used for S. Each contrast step lasted 2 s (5 Hz) and they were presented in two different pseudo-random orders, of which one is shown in Fig. 8A.

In the morning, the average mutual information between stimulus and response was almost exactly the same for synapses in the ON and OFF channels ($0.44 \pm 0.04$ bits s$^{-1}$ and $0.46 \pm 0.03$ bits s$^{-1}$, respectively). In the afternoon mutual information increased through both channels although the increase in OFF synapses (370%) was significantly larger than in ON (270%; KS test, $p < 0.001$; Fig. 8B). In OFF synapses, the maximum mutual information of 2.1 bits s$^{-1}$ was associated with an average release rate of 2.5 vesicles s$^{-1}$ around $C_{1/2}$ (Fig. 2C), equivalent to an efficiency of ~0.8 bits per vesicle.

Several of the synaptic properties we have analyzed will contribute to the improvement in information transmission in the afternoon, including the increase in synaptic gain (Fig. 2), the decrease in spontaneous noise (Fig. 3) and reduced variability of stimulus-evoked responses (Figs. 5 and 6). All these processes were subject to modulation by dopamine and, consistent with these, the mutual information in the morning was increased by activation of D1 receptors while in the afternoon it was decreased by antagonizing the effects of endogenous dopamine (Fig. 8B). Antagonizing D1 receptors did not, however, reduce mutual information to levels measured in the morning, leaving open the possibility that other signaling pathways or other neuromodulators also contribute to diurnal changes in the signaling of temporal contrast.

**Changes in the efficiency of the vesicle code**. The transfer of information through neural circuits using spikes and vesicles is the major consumer of energy in the brain with one estimate being of the order of ~24,000 ATP molecules per bit[42,43]. The largest part of this energy consumption is taken up by synaptic transmission so a key question becomes the effect of

neuromodulation on the efficiency with which vesicles are used. Strikingly, the 2.7-fold increase in information transmitted through ON synapses in the afternoon (Fig. 8B) was *not* associated with any change in the average rate of vesicle release (Fig. 2D, E), while the 3.7-fold increase in OFF synapses was associated with only a 2-fold increase in the rate around $C_{1/2}$ (Fig. 2C, E). The diurnal increase in synaptic gain was therefore associated with a 1.4- to 2.7-fold increase in the average efficiency with which vesicles were used to encode changes in contrast.

How is this increase in the efficiency of the vesicle code achieved? The simple comparison of information transmission with average rates of vesicle release obscures the fact that changes in contrast cause changes in both the rate and amplitude of release events[19]. This hybrid coding strategy is significant because the distribution of MVR events is also a function of *Zeitgeber* time and larger events are rarer and carry more specific information, as shown in Fig. 8C, D. Averaging across ON and OFF channels and times of day, univesicular events carried $0.25 \pm 0.03$ bits while events comprising 6 vesicles contained $1.03 \pm 0.14$ bits (equivalent to 0.17 bits per vesicle). Diurnal changes in the efficiency of the vesicle code therefore depend on the electrical and biochemical processes that control the fusion of vesicles in the synaptic compartment.

## Discussion

The plasticity of synapses allows the flow of information through circuits to be modulated[1] and this study provides a quantitative understanding of this idea in the context of the diurnal control of visual processing in the retina. The physiology of this circuit is regulated by circadian clocks intrinsic to photoreceptors and neurons in the inner retina and these become entrained by light and dopamine[7,44–46]. We find that this cycle alters the transmission of visual information through bipolar cells by factors of ~4 during daylight hours by adjusting four synaptic properties; the number of vesicles released by a stimulus (Fig. 3), spontaneous synaptic noise (Fig. 4), the variability of stimulus-driven responses (Figs. 5 and 6) and the balance between univesicular and multivesicular release (Fig. 7). Crucially, the shift towards MVR events of larger amplitude also increases the amount of information transmitted per vesicle (Fig. 8). Dopamine plays a major role in regulating all these aspects of retinal function although the relative contributions of these mechanisms differed between ON and OFF pathways. In the ON pathway, for instance, the reduced variability of synaptic responses and increased emphasis on MVR increased information transfer *without* an increase in synaptic gain or decrease in spontaneous noise.

**Diurnal modulation of gain**. Dopamine-dependent changes in the synaptic gain of bipolar cells might be caused by direct modulation of processes within the terminal compartment and/or actions on the circuitry in which they are embedded. Evidence for direct actions is provided by electrophysiological experiments in bipolar cells isolated from the retina of goldfish which show that activation of D1Rs potentiates the L-type calcium channels that control vesicle release[4]. We now need to improve our understanding of which types of bipolar cell are under the strongest modulatory control. In mice, D1Rs are found on the terminal compartment of bipolar cells driven by cones[47,48] but in zebrafish we only know that D1 receptors identified by in situ hybridization are found in most types of retinal neuron[49]. We also need to understand whether calcium channels are the only target of dopamine on bipolar cell terminals because capacitance measurements also demonstrate diurnal modulation of processes downstream of the calcium signal: the efficiency with which calcium triggers vesicle release is higher during the day compared

to night, reflecting higher numbers of synaptic ribbons attached to an active zone[50].

Beyond the terminal itself, dopamine receptors of different types are found on most major classes of retinal neuron, where they can modulate both chemical and electrical synapses to adjust the luminance sensitivity of the retinal circuit[8]. Less clear is how dopamine acting at these other targets might be involved in diurnal control of contrast sensitivity. Activation of D2 receptors on cone synapses can potentiate the visual drive to bipolar cells but this mechanism alone does not easily explain the *transient* increase in contrast sensitivity in the afternoon given that luminance sensitivity, a much more direct reflection of the strength of cone output, gradually increases throughout the day (cf. Fig. 1B, E).

The output from bipolar cells is also strongly dependent on the inhibitory signals that the synaptic compartments receive from amacrine cells. In mice, the activation of D1 receptors reduces inhibitory inputs that OFF cone bipolar cells receive from narrow-field glycinergic amacrine cells[51], while D1 receptors on a subset of wide-field GABAergic amacrine cells reduce inhibitory inputs onto the terminals of rod bipolar cells[52]. The possibility of diurnal modulation of inhibition remains open.

There is a good possibility that neuromodulators other than dopamine will also act on the synaptic output of bipolar cells, either directly or indirectly, to regulate the visual signal transmitted to ganglion cells. Amacrine cells release a number of neuroactive substances, including melatonin[53], Substance P[54] and somatostatin[55] and some of these can antagonize the actions of others[54]. A large number of different proteins control the activity of the retinal circuit and 17% of genes in zebrafish are under circadian regulation[7].

**Diurnal modulation of variability**. It has long been appreciated that synaptic noise can reduce the amount of information transmitted through a circuit of neurons[32]. When the retina operates under photopic conditions, for instance, the release of vesicles from bipolar cells adds noise to the signal arriving from cones causing a loss of information in RGCs[56]. It has been suggested that under other circumstances the noise in synaptic transmission might *improve* information transmission, such as when stochastic resonance increases the probability of post-synaptic depolarization crossing threshold for spike generation[33,57]. It seems unlikely, however, that the retina of zebrafish operates under such a regime, given that diurnal increases in synaptic gain went hand-in-hand with a *reduction* in several sources of noise, including spontaneous release unrelated to a stimulus.

The variability in the synaptic response of bipolar cells was quantified as average Fano factors of 1.7–2.6 (Fig. 5B–E) but the spike responses of post-synaptic ganglion cells have been reported to be more reliable, occurring with Fano factors varying from 1.5 at low contrasts down to 0.3 at higher contrasts[34]. The lower variability of responses in RGCs is not unexpected given that these neurons integrate signals from multiple bipolar cell synapses, but we now need to understand how far noise at this stage of the visual system is also diurnal control. Ultimately, the variability of signals leaving the retina should be compared with the variability of the behavioural responses they drive.

The variability of neural responses is a key determinant of the amount of information they can carry. Vesicles released individually contained 0.25 bits of information (Fig. 8C, D). A comparison can be made with the information transmitted by spikes in RGCs, where the most sluggish cells transmit ~3.5 bits/spike, while those that fire most briskly encode ~2 bits/spike[58]. A spike in an RGC therefore contains about ten times as much

information about a visual stimulus as a glutamatergic vesicle released from a bipolar cell, although this ratio is likely to vary between different functional types of RGC. It is notable that an increase in the spike rate of an RGC is associated with a *decrease* in the amount of information per spike[58] while here we find that an increase in the rate of vesicle release from bipolar cells is associated with an *increase* in information per vesicle.

All the aspects of synaptic function that we compared in the morning and afternoon were sensitive to the activation of D1 receptors, indicating that dopamine adjusts information transmission by orchestrating changes in both the signal and the various noise sources that cause it to vary. The balance between modulation of signal and noise was, however, strikingly different in the ON channel, where synaptic gain was *not* under diurnal modulation, compared to the OFF channel, where both signal and noise were regulated. Why such differential modulation of ON and OFF channels? The reasons are unclear at present, but may be related to the distributions of positive and negative contrasts in the watery environment. Regardless of our ability to answer the "Why?" question, we need to understand the processes by which dopamine and other neuromodulators adjust synaptic noise. These are likely to involve both direct actions on the synaptic compartment and indirect actions on other components of the retinal circuit. For instance, inhibitory inputs from amacrine cells inject noise into the terminal of bipolar cells[59] and it may be that dopamine receptors on amacrine cells modulate this source of variability.

### Modulation of multivesicular release

MVR is not just a property of ribbon synapses but is also a feature of synaptic transmission in the hippocampus[60], cerebellum[61] and somatosensory cortex[21], where arrival of a spike can often trigger release of two or more vesicles at an active zone. A recent combination of electrophysiology with correlative light-and electron-microscopy has even led to the suggestion that MVR may be a fundamental mode of synaptic transmission throughout the nervous system[22]. It is also recognized that MVR can be adjusted by neuromodulation, for instance through muscarinic acetylcholine receptors in the striatum[62] or GABA$_B$ receptors in the cortex[63], although the implications for information transmission in these contexts are not known. Our study has demonstrated that potentiation of MVR in the retina not only increases the amount of information that a synapse can transmit but also the efficiency of the vesicle code. It will be interesting to establish how far neuromodulators acting in other parts of the brain alter the efficiency of the information transmission and whether this involves modulation of gain as compared to the noise that is a prominent feature of many central synapses[32,33].

In the future, we will need to understand not just how the information carried by vesicles is modulated, but also how this impacts on the information contained in the spikes generated postsynaptically. How do changes in the vesicle code affect the spike code?

## Methods

### Zebrafish husbandry

Fish were raised and maintained under standard conditions on a 14 h light/10 h dark cycle[39]. To aid imaging, fish were heterozygous or homozygous for the casper mutation which results in hypopigmentation and they were additionally treated with 1-phenyl-2-thiourea (200 μM final concentration; Sigma) from 10 hours post-fertilization (hpf) to reduce pigmentation. All animal procedures were performed in accordance with the Animal Act 1986 and the UK Home Office guidelines and with the approval of the University of Sussex Animal Welfare and Ethical Review Board. The composition of the E2 medium in which experiments were carried out was as follows: Na$_2$HPO$_4$ 0.05 mM, MgSO$_4$ 1 mM, KH$_2$PO$_4$ 0.15 mM, KCl 0.5 mM, NaCl 15 mM. CaCl 1 mM, NaHCO$_3$ 0.7 mM, pH 7.0–7.5.

### Transgenic fish

Experiments were carried out using the following transgenic lines of zebrafish:

(i) *Tg(ribeye:Zf-SyGCaMP2)* expressing the synaptically localized fluorescent calcium reporter SyGCaMP 2.0 in retinal bipolar cells under the ribeye-A promoter[26].

(ii) *Tg(−1.8ctbp2:Gal4VP16_BH)* fish that drive the expression of the transcriptional activator protein Gal4VP16 were generated by co-injection of I-SceI meganuclease and endofree purified plasmid into wild-type zebrafish with a mixed genetic background. A myocardium-specific promoter that drives the expression of mCherry protein was additionally cloned into the plasmid to allow for phenotypical screening of founder fish.

(iii) *Tg(10xUAS:iGluSnFR_MH)* fish driving the expression of the glutamate sensor iGluSnFR under the regulatory control of the 10 x UAS enhancer elements were generated by co-injection of purified plasmid and tol2 transposase RNA into offspring of AB wild-type fish outcrossed to casper wild-type fish. The sequences for the myocardium-specific promoter driving the expression of enhanced green fluorescent protein (mossy heart) were added to the plasmid to facilitate the screening process.

(iv) *Tg(−1.8ctbp2:SyGCaMP6)* fish were generated by co-injection of I-SceI meganuclease and endofree purified plasmid into wild-type zebrafish with a mixed genetic background. The GCaMP6f variant was kindly provided by L. Looger (Janelia Farm). This variant holds a T383S mutation in comparison to the commercially available GCaMP6-fast version (Addgene plasmid 40755).

(v) *Tg(isl2b:nlsTrpR, tUAS:memGCaMP6f)* which drives the expression of memGCaMP6f in the optic tectum was generated by co-injecting pTol2-isl2b-hlsTrpR-pA and pBH-tUAS-memGaMP6f-pA plasmids into single-cell stage eggs. Injected fish were out-crossed with wild-type fish to screen for founders.

### Multiphoton imaging in vivo

Experiments were carried out on a total of 117 zebrafish larvae (7–9 days post-fertilization). Fish were immobilized in 3% low melting point agarose (Biogene) in E2 medium on a glass coverslip (0 thickness) and mounted in a chamber where they were superfused with E2. Imaging was carried out using a two-photon microscope (Scientifica) equipped with a mode-locked titanium-sapphire laser (Chameleon, Coherent) tuned to 915 nm and an Olympus XLUMPlanFI 20x water immersion objective (NA 0.95). To prevent eye movements, the ocular muscles were paralyzed by injection of 1 nL of α-bungarotoxin (2 mg/mL) behind the eye. Most imaging was carried out in the dorsal retina.

The signal-to-noise ratio of the microscope was optimized by collecting photons through both the objective and a sub-stage oil condenser (Olympus, NA 1.4). Emission was filtered through GFP filters (HQ 535/50, Chroma Technology) before detection with GaAsP photomultipliers (H7422P-40, Hamamatsu). The signal from each detector passed through a current-to-voltage converter and then the two signals were added by a summing amplifier before digitization. Scanning and image acquisition were controlled under ScanImage v.3.6 software[64]. In iGluSnFR recordings images were acquired at 10 Hz (128 × 100 pixels per frame, 1 ms per line) while linescans were acquired at 1 kHz. In GCaMP recordings images were acquired at 20 Hz (128 × 50 pixels per frame, 1 ms per line). Full-field light stimuli were generated by an amber LED ($l_{max}$ = 590 nm, Thorlabs), filtered through a 590/10 nm BP filter (Thorlabs), and delivered through a light guide placed close to the eye of the fish. These wavelengths will most effectively stimulate red and green cones but do not stimulate UV cones which are enriched in the strike zone[65]. The microscope was synchronized to visual stimulation. Relative changes in fluorescence (ΔF/F) during a stimulus were measured relative to the baseline before the stimulus.

### Sample size and data exclusion

No statistical methods were used to predetermine sample sizes. Experiments were repeated until trends in results were clear and this resulted in sample sizes at least equivalent to previous publications. Different experiments were repeated between 10 and 50 times and only reported if similar results were observed in >95%.

Data were only excluded from the analysis if the signal-to-noise ratio (SNR) of the iGluSnFR signals elicited at a given synapse was not sufficient to detect unitary responses to visual stimuli with a SNR of at least three.

### Stimulation protocols

Only full-field stimuli were used in which light intensity was varied sinusoidally over time i.e we only explored the signalling of temporal contrast. There was no variation in intensity over space and the stimuli did not therefore contain any spatial contrast. Measurements of contrast sensitivity with SyGCaMP2 were made by stimulating the fish with a series of 10 s stimuli (modulation at 5 Hz) around a mean intensity of 55 nW mm$^{-2}$. Measurements of contrast sensitivity with iGluSnFR used 2 s stimuli. To measure the distribution of events amplitudes and the temporal precision fish were continuously stimulated for 30 s at a given contrast.

Luminance sensitivity was assessed by stimulating the fish with a series of light steps (4 ×3 s) at 9 different light intensities increasing in steps of 0.5 log unit steps ranging from 11 pW mm$^{-2}$ to 110 nW mm$^{-2}$ (equivalent to 3.3 × 10$^{11}$ photons mm$^{-2}$).

**Drug injections**. Dopamine signalling was manipulated by injecting the antagonist of D1 receptors SCH 23390 at a final estimated concentration of 20 nM (Sigma). Finally, the long-lasting dopamine receptor ligand [3H] 2-amino-6,7-dihydroxy 1,2,3,4-tetrahydronapthalene (ADTN) (Sigma) was injected to a final estimated concentration of 200 nM. We confirmed that these drugs gained access by including 1 mM Alexa 594 in the injection needle; within 5 min of injection the dye could be detected within the inner plexiform layer of the retina. Vehicle injection did not affect synaptic responses to varying contrast.

**Calculation of temporal jitter**. In order to quantify variability in the timing of glutamatergic events, we first calculated the vector strength, $r_q$, for events composed of $q$ quanta:

$$r_q = \frac{1}{N_q} \sqrt{\left(\sum_{i=1}^{N_q} \cos\left(\frac{2\pi t_{q_i}}{T}\right)\right)^2 + \left(\sum_{i=1}^{N_q} \sin\left(\frac{2\pi t_{q_i}}{T}\right)\right)^2} \quad (1)$$

where $t_{qi}$ is the time of the $i$th $q$ quantal event, $T$ is the stimulus period, and $N_q$ is the total number of events composed of $q$ quanta. The method for estimating $t$ and $q$ is summarized in Supplementary Fig. 4. The temporal jitter, $J_q$, can then be calculated as

$$J_q = \frac{\sqrt{2(1 - r_q)}}{2\pi f} \quad (2)$$

where $f$ is the stimulus frequency.

**Calculations based on information theory**. To quantify the amount of information about a visual stimulus that is contained within the sequence of release events from an active zone we first needed to convert bipolar cell outputs into a probabilistic framework from which we could evaluate the specific information ($I_2$), a metric that quantifies how much information about one random variable is conveyed by the observation a specific symbol of another random variable[40]. The time series of quantal events was converted into a probability distribution by dividing into time bins of 20 ms, such that each bin contained either zero events or one event of an integer amplitude. We then counted the number of bins containing events of amplitude 1, or 2, or 3, etc. By dividing the number of bins of each type by the total number of bins for each different stimulus, we obtained the conditional distribution of **Q** given **S**, $p(Q \mid S)$, where **Q** is the random variable representing the *quanta/bin* and **S** is the random variable representing the *stimulus contrasts* presented throughout the course of the experiment. In the absence of information about the distribution of contrasts normally experienced by a larval zebrafish, a uniform distribution of contrasts was used for S. Each contrast step lasted 2 s (5 Hz) and they were presented in two different pseudo-random orders, of which one is shown in Fig. 8A. The contrast sensitivity varied between synapses and between morning and afternoon (Fig. 1E–G) so to make allowance for this the stimulus set S was adjusted for each synapse to span contrasts ±10% around $C_{1/2}$ measured within that synapse.

We computed the joint probability distribution by the chain rule for probability (given the experimentally defined uniform distribution of stimuli **S**):

$$p(S, Q) = p(Q|S)\, p(S) \quad (3)$$

In order to convert this distribution into the conditional distribution of S given Q, we used the definition of the conditional distribution:

$$p(S|Q) = \frac{p(S, Q)}{p(Q)} \quad (4)$$

From these distributions we computed two metrics: the mutual information I(**S**;**Q**)[66] and specific information $I_2$(**S**;**q**)[41]. Mutual information is defined traditionally as:

$$I(S;Q) = H(S) - H(S|Q) \quad (5)$$

$$I(S;Q) = \sum_{s \in S} \sum_{q \in Q} p(s, q) \log_2 \frac{p(s) \cdot p(q)}{p(s, q)} = I(Q;S) \quad (6)$$

The specific information, $I_2$(**S**;**q**), is defined as the difference between the entropy of the stimulus S minus the conditional entropy of the stimulus given the observed symbol in the response q:

$$I_2(S, q) = H(S) - H(S|q) \quad (7)$$

$$I_2(S, q) = -\sum_{s \in S} p(s) \log p(s) + \sum_{s \in S} p(s|q) \log p(s|q) \quad (8)$$

representing the amount of information observing each quantal event type q ∈ **Q** carries about the stimulus distribution **S**. Note that mutual information can also be computed from the specific information as the dot product of the specific information vector $I_2$ and the vector describing the probability of an event of a given quantal size $p(q)$. This adds to the interpretability of both metrics—the specific information is the amount of information a single (specific) symbol gives about the stimulus, and the mutual information is the average amount of information about the stimulus gained from observing any symbol.

Measuring entropy and mutual information from neural responses can be a challenging problem. Estimates require sampling from an unknown discrete probability distribution, and in many cases recording sufficient samples to observe all non-zero probability events is neither tractable nor practical. The biases introduced by undersampling can be a particular problem when the full support of the distribution (all values that map to non-zero probabilities) is high. Within the past few decades, various approaches to correcting biases in information theoretic analyses have been developed[67]. However, as the distributions of interest in this work have both a small support and are well sampled, we have opted to use standard estimates for the quantities of interest.

**Statistics**. All data are given as mean ± s.e.m. unless otherwise stated in the figure legends. All statistical tests met appropriate assumptions and were calculated using inbuilt functions in IgorPro (Wavemetrics). When data were not normally distributed we used non-parametric tests. Significance was defined as $p < 0.05$. Data collection was not randomized because all experiments were carried out within one set of animals. Delivery of different stimuli was randomized where appropriate.

**Reporting summary**. Further information on research design is available in the Nature Research Reporting Summary linked to this article.

## Data availability
Source data for all figures are provided with this paper. The datasets generated during and/or analysed as part of the current study are available from the corresponding author on reasonable request. Source data are provided with this paper.

## Code availability
The code used to analyze the data in this study is available at https://github.com/lagnadoLab/glueSniffer.

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

## Acknowledgements

The authors express many thanks to all the members of Lagnado laboratory for discussion. We also thank Tom Baden and Sylvia Schröder for their criticisms and suggestions. This work was supported by grants to L.L. from the Wellcome Trust (102905/Z/13/Z and 221936/Z/20/Z).

## Author contributions

J.M.-D. conceived, designed and executed experiments, analyzed results and prepared the manuscript. B.J. carried out analysis and wrote code. F.E. conceived and executed experiments and carried out analysis. J.J. conceived and executed experiments and carried out analysis. L.L. conceived the project, designed experiments, analyzed data, repaired equipment, wrote code and prepared the manuscript.

## Competing interests

The authors declare no competing interests.
