## [Peer review file · Nature Communications]

REVIEWER COMMENTS

Reviewer #1 (Remarks to the Author):

In this manuscript, the authors have examined how the information transfer is modulated at bipolar cell output synapses (to Amacrine and ganglion cells) during the diurnal cycle. In the first set of experiments, the authors have examined presynaptic Ca levels as an indicator of presynaptic activity, which have peaks in the afternoon. This may indicate that the diurnal modulation already starts before (?) bipolar cell synapses. In the second set of experiments, they have analyzed transmitter release using a glutamate sensor, and examined the effects of the diurnal cycle. Multi-vesicular release, spontaneous activity, and synaptic gain all contribute to the coding of visual information transfer through bipolar cell synapses. Moreover, the diurnal cycles are likely to be mediated by dopamine released from amacrine cell, though the targets of dopamine could be multiple and it is hard to identify. The study uses reliable and sophisticated optical techniques as well as quantitative analysis of information transfer, and provides very interesting information on how retinal synapses convey visual information particularly during the diurnal cycle. Although I feel the paper is highly interesting to neuroscientists outside the immediate field such as synaptic neurobiologists and neuroscientists in general, I have several issues on the manuscript regarding mechanistic insights of dopamine modulation. In short, there are so many targets of dopamine in retina, and it is unclear if the observations here are due to synaptic mechanisms or indirect effects.

(1) Presynaptic Ca levels are measured by long light stimulation, whereas transmitter release is monitored by sinusoidal stimulation. Because two readouts are measured by two different experimental protocols, it is hard to know the transfer function of the bipolar cell synapse. Perhaps this is due to low signal to noise ratio of the Ca probes. If it is doable, Ca measurements using sinusoidal stimulation will be helpful. In other words, it is not clear if dopamine modulates bipolar cell output synapses or somewhere else. It is informative to derive transfer function of the synapse and examine if dopamine modulates multi-vesicular release etc. Without knowing this, I am not certain if neuromodulators increase “synaptic transfer of information and the efficiency of vesicle code”, as the authors state.

(2) It is known that there are many targets of dopamine in the retina, and it is unclear where the modulation takes place. For example, feedback(feedforward) inhibition from horizontal cells and feedback inhibition from amacrine cells are known to be modulated by dopamine. It is very informative to examine at least a few candidates and dissect which pathway modulates which element of dopamine modulation. At least, previous works should be cited properly to discuss potential targets of dopamine. Although I understand this is difficult to address compared to the photoreceptor output synapses, understanding the specific role of the synapses in the inner plexiform layer is important and will increase the implications of the study.

(3) Hull et al. (2008) have examined the consequence of the diurnal cycle on the transmitter release process at retinal bipolar cells. It has been often observed that synaptic ribbons can be modulate during the day cycle. At least this point should be discussed in the discussion part.

Minor

(4) Although detecting transmitter release is very interesting, the postsynaptic role should be discussed somewhere.

(5) Figure 1 may be reanalyzed so that one can see the differences between ON and OFF bipolar cells. It looks inconsistent with other figures.

Reviewer #2 (Remarks to the Author):

Summary: The authors present an interesting and well-done study, using novel measurement and analysis techniques, to investigate circadian modulation of synapses in the retina. They found that dopamine is a crucial modulator at the bipolar cell synapses and found interesting differences between the ON and OFF retinal bipolar cell pathways. Especially interesting, they were able to calculate that the amount of information transmitted by the bipolar synapse varies in a circadian way.

There are a few points detailed below that need to be addressed for the strength of the paper.

Major points:

1) Figure 1. Please give a rationale for why the contrast sensitivity was tested with sinusoidal variations. Did this give a larger response? Also explain why these are called temporal contrasts in line 118, which is presumably related. For the optic tectum responses there are clearly separate responses to each variation, but this is less clear in the BC responses in Figure 1. Does the change in fluorescence measurement relate to the total change from baseline, or the average change in the peaks that oscillate around the mean luminance?

2) Figs 2-3 Can you comment on how similar the responses between different synapses were? Were different types of ON and OFF BCs sensed by the IgluSNfr? It seems that the effect of D1R agonists/antagonists might vary between BC types. In the mammalian retina at least there are specific BC types that lack D1Rs (e.g. Farshi P, Fyk-Kolodziej B, Krolewski DM, Walker PD, and Ichinose T. Dopamine D1 receptor expression is bipolar cell type-specific in the mouse retina. J Comp Neurol 524:

2059-2079, 2016.), although I am not aware of data for zebrafish. Given the large number of different BC types, n's of only 5-12 may not be enough to get a representation of all BC types.

3) As the authors note in the discussion, BC synapses receive significant amounts of inhibition. Given that many amacrine cell types also express D1Rs, could the authors comment on what the roles of inhibition might be here? It seems likely to play a significant role.

4) Fig 5 – does a decreased Fano factor correlate to reliability of the synapse? (line 268). That is how the author's use the factor, but it was not clear if they are directly related.

Minor points:

1) Line 59 – it is not clear what is meant by “the symbols conveying the message” This terminology is used several more times, so it would be good to define it.

2) Figure 2 – In the results it is stated that both the morning and afternoon time frames are 2 hours, but in the figure legends it says morning is Z1-3 and afternoon is Z6-9. Please correct the discrepancy. Also please clarify here what quanta/cycle means (is this cycle of the sine wave variation in contrast?).

3) Figure 3 – please make a clearer visual distinction between the two control responses in Fig 3 B and C and add them to the figure legend. It is critical to be able to compare the responses before and after agonist/antagonist application.

4) Line 268-269 Can you rephrase this sentence – I'm not sure what it is trying to say: “Notably, the variability of synaptic output was higher than expected for a Poisson process, for which the Fano factor is one.”

Reviewer #3 (Remarks to the Author):

Summary

The paper is extremely well structured and a delight to read. It has three basic parts. 1) Neural response amplitudes in bipolar cells of the inner retina of zebrafish larvae are strongly circadian, with the largest responses occurring in the early afternoon. The reviewer doesn't know that this is elsewhere published, and the finding would strongly influence the way Neurophysiological studies are conducted in the future. 2) Bipolar-cell glutamate release is circadian. There are 4 aspects of this release identified that affect synaptic ability to convey information: i) spontaneous release rate, ii) variation in signal amplitude to fixed stimuli or Fano factor, iii) the temporal precision or ability to phase-lock the stimuli. iv) the types of symbols or tokens released in response to stimulation. 3) All these four factors controlling the

efficiency of synaptic information transfer in the inner retina are under control of the diurnal control of the neuromodulator dopamine.

Vesicle release is not Poisson with a Fano factor equal to 1.0 but typically involves Fano factors greater than 1, suggestive of vesicle clustering, and ultimately, multivesicular release. Multivesicular release of up to 5 vesicles simultaneously occurs, and the various packets of vesicles released comprise the multiple symbols of information conveyed by the synapse. The 5-vesicular packet is favored in the afternoon, has the least temporal jitter, and conveys the most information. This finding provides a definite role for multivesicular release.

The net impact is that one should not be looking for stable inner retinal circuits in the inner retina to process information. They vary by the hour under neuromodulator control. The net suggestion, though not explicitly stated, is that the bipolar-to-ganglion-cell synapse is the information bottleneck in the retina. The outer retina responses of bipolar cells, which may involve only a few synapses, appear quite stable, although diurnal. While there is a long history of diurnal dopaminergic synaptic modification in the outer retina, this is the first exploration of the effects of dopamine on the inner retina synapses of bipolar cells.

Minor comments:

Abstract:

Neither the abstract nor the title mentions the species.

Introduction:

-- line 42 'modulators vs modulations'. The words look too similar. Maybe substitute 'changes' for modulations.

--line 43 'Diurnal control of temporal contrast processing' Is the meaning diurnal or circadian?

There is a long history of circadian processes modulated by dopamine in the outer retina, which is passed over in the introduction (HJ Wagner, 1980).

Results:

--line 81 'Zeitgeber' The reviewer had to look this word up, perhaps other readers would also. In general it appeared to have a circadian sense (as opposed to diurnal).

--line 116 'contrast' The reviewer identifies this term with temporal sine waves over a mean luminance. These are step functions on no background.

-line 120 'C1/2 it fell' Fix grammar.

--line 134 'individual active zones vesicles' unclear 'vesicles from individual active zones'?

--line 158 'at an individual OFF active zone' 'an individual OFF-bipolar active zone'

--line 201 'The dynamic range over which D1 receptors adjusted synaptic gain' is this synaptic gain or synaptic output. The changes seen might originate in distal retina, as seen in Fig. 1.

--line 270-1 'The spike responses of post-synaptic RGCs are less variable, with a Fano factor as low as 0.3 at higher contrasts' But Berry shows a Fano factor of 1.5 at low contrasts.

Line 441 '...an increase in vesicle release rate is associated with an increase...' The 'i' in 'increase' number two needs italic font.

Methods

Line 655 'E2 medium' Specify source or composition

Line 671 'Full-field light stimuli were generated by an amber LED (I_{max} = 590 nm Thorlabs), filtered through a 590/10 nm BP filter (Thorlabs),' Then most of what is described is lws cone physiology. This maybe should be discussed. There are no UV cone signals. The experiments don't determine whether the UV pathway is dopamine sensitive, for instance. As noted 'These wavelengths will most effectively stimulate red and green cones.'

--line 688 '...200 nM (Sigma). Finally,...' typo.

--line 699 '...where t_{qi} is the time of the ith q-quantal event,...' How is t_{qi} determined?

--line 700 'number of events of composed of q-quanta.' typo

Fig. 1 legend

A. 'light step' Although the 590nm wavelength is buried in the methods, which this Journal leaves until last, it would be helpful to include the wavelength in the legend, and mention that it is the lws cone pathway that is being studied.

B. --line 95-6 '...region of showing terminals...' typo

E. define 'contrast' as used in used in this figure. The stimuli are all step functions, presumed 100% contrast.

F. --line 109 'h=7.0' Really?

--line 113 'All error bars show ± 1 SD.' Error bars are SD (not SE?)

The legend should mention the larval age.

Figure 2 legend

A. In this and succeeding figures, it isn't clear how ON and OFF responses are distinguished using sinusoidal stimuli, or is this determined in separate experiments with other stimuli?

--line 141-2 '...using a stimulus of variable contrast..' using stimuli of variable contrast

--line 158 'at an individual OFF active zone' 'an individual OFF-bipolar active zone'

Fig. 3 legend

Specify the concentration of SCH23390 in the legend.

--line 190 'Each point shows the mean ± s.e.m.' Here you give 'sem' is that also true of Fig. 1, where 'SD' was given? The methods say sem unless otherwise noted.

--line 193 C. 'synapses in three conditions: afternoon (green dots),' There are no green dots in panel 'C.'

Fig. 4 legend

In B. and C. it appears to the reviewer that event rate should be the X-axis and the Q_e should be the Y-axis. It is hard to contemplate that the Q_e determines the spontaneous event rate.

Fig. 6 legend

--line 307 '...afternoon (green dots; n = 14 synapses) during the (open green dots; n= 10 synapses) and...'
Add 'morning'

Fig. 7 legend

B, C, D, E, The symbol legends indicate that two items with different color codes are being compared, but there are 3 colors on the bar graphs: gray and two shades of either red or green. The reviewer couldn't read the bar charts.

Suppl Fig S2

Panel B. Four traces but only two colors. The trace labels could be put over individual traces rather than all in a single box.

Reviewer #4 (Remarks to the Author):

The manuscript by Moya-Diaz and colleagues describes a most interesting case of neuromodulator mediated neural network adjustment in the retina. The authors convincingly show that synapses of bipolar cells shown diurnally regulated visual processing, affecting synaptic gain, variability, and noise. The efficiency of synaptic transmission is modified by potentiation of multivesicular release. These new results are satisfyingly linked to the large literature on the role of the neuromodulator dopamine in retinal plasticity.

The results are well presented and the conclusion are both supported by the data and novel. Overall this is a fantastic paper using an original experimental approach to settle an interesting question.

However, before the study can be published there are many points that need to be addressed, many of them minor, but also a couple major ones as detailed in the following:

Major points:

1. There are major problems with Figure 1:

In Figure 1 BCD, as it is indicated in the legend, the data from B is plotted in C. However there must be a mix-up, including a wrong color label. Although the y axis is the same, the red line in C is probably okay, however the blue line in C should be black. It is not clear what data is plotted by the black line in panel C and certainly absent from panel B. Information in panel D plots data from panel C, but here the sensitivity is highest at ZT16, which is not possible according to B and according to the fact that they larvae are more or less blind at ZT16.

Also in D, I am really wondering if the y axis is scaled as in the figure (0.1, 1,10,100), what scale will be taken for the error bars? The reason I ask is that the error bars looked extraordinarily big in this figure.

Hence Figure 1 must be reworked and corrected.

2. Figure 2 is at best misleading:

I am confused on how the data depicted in panel E can be squared with the data depicted in B and C. The authors should check their plot, and if correct explain panel E better.

(For instance, I am confused how the similar response at 20% contrast in panel C relates to the very different values in panel E. It may be my lack of understanding, but then a clearer explanation is warranted.)

Minor points:

Line 43-44: "focusing on the visual signal transmitted that bipolar cells transmit to the inner retina."
Delete "transmitted"

Line 63 "instead composed of a number of symbols, composed of one, two, three or more" replace one "composed"

Line 82, "larvae are blind at subjective night" The term "subjective" is used by conventional chronobiologists exclusively when referring to time without any zeitgebers. Delete "subjective".

Line 86.87 "Over the course of the day, luminance-sensitivity increased gradually over a range greater than 200-fold (Fig. 1D)." this may need to be edited because of the points mentioned related to the figure BCD.

Line 95, ensure abbreviations like "IPL" are at least spelled out once at first appearance in the text.

Line 103, "red arrow" should be "dashed red arrow".

Line 106, the sample size should be indicated and the ON or OFF terminals should be specified.

Line 134-136 "Synaptic function was compared over a two-hour period beginning 1 hour after light onset ("morning") with a two-hour period beginning 6 hours later ("afternoon"; Fig. 1G). This reads slightly confusing. It is better to specify the recording period in Zeitgeber time, like ZT1-ZT3,ZT7-ZT9

Line 143 "ZT 6-9 hours", probably ZT7-ZT9 is meant – please verify.

Line 145 The legend of figure 2C should contain some information on the statistical treatment

Line 155, what do the error bars represent in panels C and D? Statistical information missing?

Line 168-170, "Contrasts in natural visual scenes rarely exceed 40% and in the morning this range was signalled best through the ON channel. But in the afternoon the OFF channel became dominant, with contrast gains increasing by factors of 2-6." This conclusion may need to be clarify a bit better, including the value of 2-6 needs to be more clearly specified.

Line 193, "in ON bipolar cell synapses in three conditions: afternoon (green dots)" - there are no green dots. "afternoon (green dots)" should be deleted. "three" to "two"

Line 204-205, "But diurnal modulation of gain was narrower than this potential range: 1.7-fold in OFF synapses and 1.1-fold in ON." But there is no diurnal modulation at ON response according to line 163" There was little diurnal modulation of the CRF measured at ON synapses" and the statistics information in the legend of Figure 2 D. But if read line 204-205 alone, it sounds there is a significantly difference between morning and evening in ON response, but the difference is very small (the ratio is only 1.1).

Line 245, what do the error bars mean in B and C? Any statistics information?

Line 266, KS test stands for Kolmogorov-Smirnov test

Line 266-267, "The increase in contrast gain and sensitivity in the afternoon (Fig. 2C-D)" in D there is no increase in the afternoon for ON response.

Figure 5: Panels D and E do not add much to panels B and C. It looks more convincing, but simple because they ignore the high variability in B and C by averaging the values again.

Line 281 Specify the meaning of the error bars in panels B and C

Line 287-288 t-test should not be used for 3 groups.

Line 290 Figure 6: The temporal precision of MVR is under diurnal control in the OFF channel” But this figure is not only about MVR.

Line 297, legend for B: what does the dash line mean?

Line 306-310, legend for C: what does the dash line mean?

“The relationship observed in the morning is better described by a straight line with a = 34.7 ± 1.5 and a slope = -3.6 ± 0.5 .” there is no fitting in the figure for morning data, but afternoon data and morning +D1 agonist data.

Line 322, (Fig. 3H; t-test at each Qe).I guess it meant Figure 6 C and t test should not be use with 3 groups in the figure.

Line 325-326 “Diurnal variations in dopamine therefore modulate the temporal accuracy of vesicle release” should add “for OFF synapses”

Line 338 Legend for A, is this an OFF or an ON terminal?

Line 340-347 legend B-E n should be specified.

Line 342 legend for Figure 7B “($p < 0.059$, Chi-squared test)” and line 352 “(Fig. 7B; $p < 0.05$, KS test)” does not fit.

Line 346 legend for D “($p < 0.007$)” and line 354 “(Fig. 7D; $p < 0.02$)” does not fit

Line 359-361 “Qualitatively similar modulation of MVR was observed over a range of contrasts from 20% to 80% and blocking D1 receptors in the afternoon shifted the distribution back to univesicular release in both ON and OFF channels (Fig. 7D and E).” The data is not shown for other contrasts, so it is needed to be indicated as “data not shown”. Furthermore, the second half of the sentence sounds for me that the effect of blocking D1 receptors at other contrast levels have also been checked. If this is the case, “data not shown” can be added here, or if not, “at 60% contrast” should be added.

Line 361-362, “Diurnal variations in dopamine therefore modulate MVR.” I think should be “Diurnal variations in dopamine therefore modulate the probability or distribution of MVR”

Line 365 “How do changes in synaptic gain (Figs. 2-3), noise (Fig. 4-6) and MVR (Fig. 7)” noise is only investigated in figure 4.

Line 392 “(270%; $p < 0.001$; Fig. 8B)” which test?

Line 394 “release rate of 2.5 vesicles s⁻¹ around C1/2,” Where does this value come from?

Line 415, legend for Figure B, what do the error bars represent? Sample size? Any statistics information?

Line 417-419 legend for C and D, for each figure, there is only one sample size provided. Does it apply to both recordings in the afternoon and morning? Is the curve fitted according to morning values or afternoon values? What do the error bars signify?

Line 472-474 “Most strikingly, reduced variability of synaptic responses and increased emphasis on MVR increased information transfer through the ON pathway without an increase in synaptic gain.” This reads strange and should be reworded.

Some additional recent references on the effect of circadian changes on vision in the discussion would also improve the manuscript a bit.

Besides these many small issue, this is an excellent study.

Response to Reviewers

Reviewer 1

Thank you for your helpful and positive comments

In this manuscript, the authors have examined how the information transfer is modulated at bipolar cell output synapses (to Amacrine and ganglion cells) during the diurnal cycle. In the first set of experiments, the authors have examined presynaptic Ca levels as an indicator of presynaptic activity, which have peaks in the afternoon. This may indicate that the diurnal modulation already starts before (?) bipolar cell synapses. In the second set of experiments, they have analyzed transmitter release using a glutamate sensor, and examined the effects of the diurnal cycle. Multi-vesicular release, spontaneous activity, and synaptic gain all contribute to the coding of visual information transfer through bipolar cell synapses. Moreover, the diurnal cycles are likely to be mediated by dopamine released from amacrine cell, though the targets of dopamine could be multiple and it is hard to identify. The study uses reliable and sophisticated optical techniques as well as quantitative analysis of information transfer, and provides very interesting information on how retinal synapses convey visual information particularly during the diurnal cycle. Although I feel the paper is highly interesting to neuroscientists outside the immediate field such as synaptic neurobiologists and neuroscientists in general, I have several issues on the manuscript regarding mechanistic insights of dopamine modulation. In short, there are so many targets of dopamine in retina, and it is unclear if the observations here are due to synaptic mechanisms or indirect effects. In other words, it is not clear if dopamine modulates bipolar cell output synapses or somewhere else.

We fully agree with the reviewer last comment and went into this issue in a subsection of the Discussion headed “Diurnal modulation of gain” which starts: “Dopamine-dependent changes in the synaptic gain of bipolar cells might be caused either by direct modulation of processes within the terminal compartment or by actions on the circuitry in which they are embedded”. There we highlighted the fact that the gain of the visual signal measured at the output of bipolar cells depends on the gain of the on synapses that drive them as well as the inhibitory inputs that they receive from amacrine cells. **We have now expanded this discussion in response to point 2 (see below).**

The precise targets at which dopamine acts is not the focus of this work. Our purpose is to understand how diurnal modulation alters transmission of the visual to the inner retina. In the same section of the Discussion we point out that neuromodulators other than dopamine might also be involved. “Dopamine release is controlled by the internal circadian clock as well as changes in luminance⁷ or the appearance of food-related odours⁴. But other neuromodulators, are also released from amacrine cells, including melatonin⁴³, Substance P⁴⁴ and somatostatin⁴⁵ and some of these can antagonize the actions of others⁴⁴. A large number of different proteins control the activity of the retinal circuit and 17% of genes in zebrafish are under circadian regulation⁷. There is therefore a good possibility that neuromodulators other than dopamine will also act on the synaptic output of bipolar cells, either directly or indirectly, to regulate the visual signal transmitted to ganglion cells”.

(1) Presynaptic Ca levels are measured by long light stimulation, whereas transmitter release is monitored by sinusoidal stimulation. Because two readouts are measured by two different experimental protocols, it is hard to know the transfer function of the bipolar cell synapse. Perhaps this is due to low signal to noise ratio of the Ca probes. If it is doable, Ca measurements using sinusoidal stimulation will be helpful.

Thanks for highlighting this lack of clarity. We did indeed use sinusoidal stimuli for measurements of both presynaptic Ca and glutamate release but did not clearly indicate this in the stimulus trace for the Ca measurement in Fig. 1E. This has now been altered to show the modulation in intensity around the mean rather than the contrast calculated from this modulation.

It is informative to derive transfer function of the synapse and examine if dopamine modulates multi-vesicular release etc. Without knowing this, I am not certain if neuromodulators increase “synaptic transfer of information and the efficiency of vesicle code”, as the authors state.

We are not confident that we understand the reviewers point here. We document how dopamine modulates multivesicular release in Fig. 7. The amount of information transferred at bipolar cell synapses is then quantified (in bits) in Fig. 8A and B. The efficiency with which information is transmitted by vesicles, quantified as bits per vesicle, is shown in Fig. 8C and D. **We believe that this one of a very few studies to convincingly apply information theory to quantify information transfer at a synapse.**

(2) It is known that there are many targets of dopamine in the retina, and it is unclear where the modulation takes place. For example, feedback(feedforward) inhibition from horizontal cells and feedback inhibition from amacrine cells are known to be modulated by dopamine. It is very informative to examine at least a few candidates and dissect which pathway modulates which element of dopamine modulation. At least, previous works should be cited properly to discuss potential targets of dopamine.

Thanks for this prompt. We have now expanded the Discussion sections “Diurnal modulation of gain” and “Diurnal modulation of noise” to go into more detail about the sites at which dopamine might act to alter the amount of visual information transmitted from the synapses of bipolar cells. In particular, we discuss how dopamine receptors on amacrine cells might

modulate inhibitory inputs to bipolar cells, adding references to work from Barnes, Eggers, Arshavsky and von Gersdorff. There is a very large literature on the various actions of dopamine in the retina, so we also point the reader to a very good recent review by Roy and Field.

Although I understand this is difficult to address compared to the photoreceptor output synapses, understanding the specific role of the synapses in the inner plexiform layer is important and will increase the implications of the study.

We agree that it would be interesting to carry out a similar study focusing on diurnal modulation of the visual signal transmitted from photoreceptors.

(3) Hull et al. (2008) have examined the consequence of the diurnal cycle on the transmitter release process at retinal bipolar cells. It has been often observed that synaptic ribbons can be modulate during the day cycle. At least this point should be discussed in the discussion part.

Thank you for this helpful prompt. We have added the following point to the Discussion section “Diurnal modulation of gain”: “*Calcium channels may not be the only target of dopamine acting directly on bipolar cell terminals because capacitance measurements also demonstrate diurnal modulation of synaptic gain downstream of the calcium signal: the efficiency with which calcium triggers vesicle release is higher during the day compared to night, reflecting higher numbers of synaptic ribbons attached to an active zone*{Hull, 2006 #253}.”

Minor

(4) Although detecting transmitter release is very interesting, the postsynaptic role should be discussed somewhere.

The modulation of glutamate release in the inner retina also determines the visual signal transmitted out of the retina by RGCs. This was demonstrated in Supplementary Figure 2, which shows diurnal changes in the visual signal that the axons of RGCs deliver to the optic tectum assayed using a calcium reporter (membrane-targeted GCaMP6f in the axons). Fig. S2 (referenced in the last line of the first sub-section of Results) also demonstrated that changes in the retinal output were dependent on D1 receptors.

(5) Figure 1 may be reanalyzed so that one can see the differences between ON and OFF bipolar cells. It looks inconsistent with other figures.

We cannot see an inconsistency with other figures. If the reviewer is referring to the traces in B and C, they are indeed very different to the traces in later figures because Fig. 1 is based

on measurements using SyGCaMP while later figures use iGluSnFR. SyGCaMP signals are slower because the underlying calcium transients are slower than glutamate transients and because the GCaMP reporter has significantly slower off-rate than iGluSnFR.

Figs.S1B and C show separate luminance-response plots for ON and OFF channels measured with SyGCaMP and the changes in these during hour of light as well as D1R-dependent changes. ON and OFF channels both show diurnal changes in luminance sensitivity.

Measurements of contrast sensitivity made with SyGCaMP (Fig. 1G) did not show significant differences between ON and OFF terminals when we averaged across each population. It is only when measuring glutamate release with the sensitivity and resolution of iGluSnFR that we observed obvious differences between ON and OFF channels, so we do not feel that it would be helpful to separately plot these functions for each class. We have added a note about this in the legend to Fig. 1.

This begs the question: Why are diurnal differences in contrast-sensitivity obvious in the OFF channel but not the ON when measured with iGluSnFR but not when measuring with SyGCaMP? It may be that this reflects the difference in the SNR of the two reporters. Another possibility (beyond the scope of this paper) is that the relation between calcium and release rate is modulated differently in ON and OFF channels. We cannot assume that the function relating calcium to release rate is fixed for ON and OFF channels throughout the light-dark cycle. This is the major reason for preferring a reporter of the actual synaptic output (iGluSnFR) for understanding changes in transmission of the visual signal to the inner retina.

Reviewer 2

Summary: The authors present an interesting and well-done study, using novel measurement and analysis techniques, to investigate circadian modulation of synapses in the retina. They found that dopamine is a crucial modulator at the bipolar cell synapses and found interesting differences between the ON and OFF retinal bipolar cell pathways. Especially interesting, they were able to calculate that the amount of information transmitted by the bipolar synapse varies in a circadian way.

We are glad that you found the paper interesting and thank you for your suggestions below.

There are a few points detailed below that need to be addressed for the strength of the paper.

Major points:

1) Figure 1. Please give a rationale for why the contrast sensitivity was tested with sinusoidal variations. Did this give a larger response? Also explain why these are called temporal contrasts in line 118, which is presumably related.

This paper only uses full-field stimuli in which light intensity is varied over time i.e we only explore the signalling of *temporal* contrast. There is no variation in intensity over space and the stimulus does not therefore contain any *spatial* contrast. This is now made clear in the beginning of the Methods subsection “Stimulation protocols”.

Temporal contrast could also have been probed investigated using stimuli in which intensity varied as, for instance, a square wave (e.g Baden et al 2013). A sinewave is generally preferred, however, because it is a “cleaner” stimulus being composed of just one frequency component. The use of a full-field stimulus that varies sinusoidally in time is a standard approach in visual neuroscience that has been widely used for measuring synaptic responses in the inner retina over many years (Ichinose et al., 2013; Baden et al 2013; James et al, 2019, Zhao et al., 2020).

For the optic tectum responses there are clearly separate responses to each variation, but this is less clear in the BC responses in Figure 1.

The difference in the shapes of these traces reflects the difference in frequency at which the stimulus was modulated. Throughout the primary paper stimuli were modulated at 5 Hz. The resulting variations in synaptic activity can be seen clearly when using iGluSnFR as the reporter of synaptic activity (Figs. 2-8) but not when using the slower calcium reporter SyGCaMP2 (Fig. 1E). It is because of the slow speed of the SyGCaMP2 reporter that we reduced the stimulus frequency to 1 Hz in Supplementary Figure 2. We now flag up this key difference in the legend to Fig. S2 with the sentence: “*Note that here the stimulus was modulated at 1 Hz (full field, sine wave) rather than the 5 Hz used in the main body of the paper*”. A secondary (but less important change) is that in Fig. S2 we used a faster reporter – GCaMP6f.

Does the change in fluorescence measurement relate to the total change from baseline, or the average change in the peaks that oscillate around the mean luminance?

“Relative changes in fluorescence ($\Delta F/F$) during a stimulus were measured relative to the baseline before the stimulus”. We have now added this sentence to the Methods sub-section “Multiphoton Imaging *In Vivo*”.

2) Figs 2-3 Can you comment on how similar the responses between different synapses were? Were different types of ON and OFF BCs sensed by the *IgluSnfr*?

If the reviewer thinking of different synapses *within the same terminal*, the comparison is best made by Fig. 1 of James et al. (2019), which is reproduced below. Different synapses from the same terminal were always strongly correlated in their activity.

Fig. 1 | Glutamate transients of varying amplitude imaged at individual active zones. a, Multiphoton section through the eye of a zebrafish larva (7 d.p.f.) expressing *iGluSnFR* in a subset of bipolar cells. **b**, Line scan through a single terminal. No other terminals were in the vicinity. **c**, The kymograph (top) shows the intensity profile along the broken red line in **b** as a function of time. The broken green trace to the side of the kymograph shows the time-averaged intensity along the line, and the red and black traces are the two Gaussians that sum to best describe this spatial profile. The amplitude of each Gaussian at each time point was used to quantify the signal at each active zone, and is plotted in the traces below. The modulation in light intensity (20% contrast, 5 Hz) is shown immediately below. Note the large variations in the amplitude of glutamate transients. **d**, Expansion of the records within the boxes outlined in broken blue lines in **c**. Sometimes both active zones release glutamate, while on other occasions only active zones 1 or 2 are active (red dotted line). Qualitatively similar *iGluSnFR* signals were observed in 187 independent experiments.

In terms of synapses from *different types of bipolar cell*, we focused on a comparison of the two most basic functional types, ON and OFF. There was significant functional heterogeneity within the ON and OFF classes of BC when we compared their contrast-response functions, and these differences are summarized below for the reviewer.

Variations in contrast-response functions (CRF) measured at the synapses of bipolar cells.

a. CRFs of different shapes could be recognized: linear, sigmoidal, saturating exponential and then others that displayed a 'dog-leg'. **b.** Average CRF in 89 OFF synapses (red) and 59 ON synapses (green), including all types shown in a. In OFF synapses the average CRF was linear up to 100% contrast but in ON synapses it saturates. **c.** The distribution of these four basic types of CRF in ON and OFF cells.

We have not added this information to the manuscript because we are not yet at the stage where we can make a systematic comparison of function and morphology. Within each functional group there were several different morphological types but our samples are not large enough to make statistically significant comparisons at this stage (as pointed out by the reviewer in the point below). But we do take the reviewers point that it would be good to give readers a better idea of the different morphological types that were included in our analysis and so the revised ms includes a new Supplementary Figure 3 that shows a sample of the different morphologies of the BCs that were labelled with iGluSnFR, as shown below.

Figure S3. Types of bipolar cell sampled in this study

A. Multiphoton section through the eye of a larval zebrafish (7 dpf) expressing iGluSnFr in a subset of bipolar cells. **B.** Examples of some of the different morphological subtypes of OFF (top) and ON (bottom) bipolar cells which

were sampled in this study. All scale bars are 2 μm . **C.** We focused on a comparison on the two most basic functional types, ON and OFF cells, identified through their responses to steps of light applied before applying stimuli modulated at 5 Hz. iGluSnFR signal were measured in a total of 91 synapses in ON and 151 synapses in OFF cells.

It seems that the effect of D1R agonists/antagonists might vary between BC types. In the mammalian retina at least there are specific BC types that lack D1Rs (e.g. Farshi P, Fyk-Kolodziej B, Krolewski DM, Walker PD, and Ichinose T. Dopamine D1 receptor expression is bipolar cell type-specific in the mouse retina. J Comp Neurol 524: 2059-2079, 2016.), although I am not aware of data for zebrafish. Given the large number of different BC types, n's of only 5-12 may not be enough to get a representation of all BC types.

We agree with the reviewer that it would not be at all surprising if D1R agonists and antagonists have differential effects on different BC types because the receptors are not expressed on all. As the reviewer points out, our sample sizes are not large enough to make clear conclusions about how the functional diversity described above and in the paper correlates with morphological diversity and receptor distribution. Previous studies in adult zebrafish described 7 subtypes of OFF cells, 6 subtypes of ON cells and 4 subtypes of bistratified cells with axon boutons or branches in the OFF and ON sublaminae (Connaughton and Nelson, 2004). The total number of BCs described in adult zebrafish is therefore at least 17.

Thank you for prompting us about the paper by Farshi et al. (2016), which provides really interesting data about the subtypes of BC expressing D1R in mice. We now cite this paper in the Discussion, while pointing out that we do not have similar information about D1R distributions in larval zebrafish.

“Dopamine-dependent changes in the synaptic gain of bipolar cells might be caused either by direct modulation of processes within the terminal compartment and/or by actions on the circuitry in which they are embedded. Evidence for direct actions is provided by electrophysiological experiments in bipolar cells isolated from the retina of goldfish which show that activation of D1Rs potentiate the L-type calcium channels that control vesicle release{Esposti, 2013 #13}. We are, however, lacking information on which types of bipolar cell might be under strongest modulatory control. In mice, D1Rs are found on the terminal compartment of bipolar cells driven by cones{Mora-Ferrer, 1999 #238;Farshi, 2016 #237} but in zebrafish we only know that D1 receptors identified by in situ hybridization are found in most cell types of retina {Li, 2007 #254}. Calcium channels may not be the only target of dopamine acting directly on bipolar cell terminals because capacitance measurements also demonstrate diurnal modulation of synaptic gain downstream of the calcium signal: the efficiency with which calcium triggers vesicle release is higher during the day compared to night, reflecting higher numbers of synaptic ribbons attached to an active zone{Hull, 2006 #253}.”

3) As the authors note in the discussion, BC synapses receive significant amounts of inhibition. Given that many amacrine cell types also express D1Rs, could the authors comment on what the roles of inhibition might be here? It seems likely to play a significant role.

We have added the following to the Discussion section “Diurnal modulation of gain”

“The gain of bipolar cell synapses is also strongly dependent on the inhibitory inputs that the synaptic compartments receive from amacrine cells and the possibility of diurnal modulation of inhibition remains open. In mice, the activation of D1 receptors reduces inhibitory inputs that OFF cone bipolar cells receive from narrow-field glycinergic amacrine cells{Mazade, 2019 #250}, while D1 receptors on a subset of wide-field GABAergic amacrine cells reduce inhibitory inputs onto the terminals of rod bipolar cells{Travis, 2018 #251}.”

And the following to the Discussion section “Diurnal modulation of noise”

“Inhibitory inputs from amacrine cells inject noise into the terminal of bipolar cells⁵⁹ and it will be interesting to establish how far dopamine receptors on amacrine cells modulate this source of variability.”

4) Fig 5 – does a decreased Fano factor correlate to reliability of the synapse? (line 268). That is how the author’s use the factor, but it was not clear if they are directly related.

No, Fano factor is higher for less reliable synapse. We have tried to make this clearer by rewording the beginning of the section on “ii) Variability in stimulus-evoked responses” as follows

“The reliability of neural responses measured as spikes can be assessed using the Fano factor: the ratio of the variance-to-mean of spikes counted in a fixed time-window after a repeated stimulus{Berry, 1997 #61;Churchland, 2010 #192}. We used a similar approach to assess the reliability of synapses, calculating the Fano factor by counting the number of vesicles released over each cycle of a sinusoidal stimulus (Fig. 5A). In the morning, F was ~2.6 in both ON and OFF synapses when averaged over a range of contrasts, but synapses were more reliable in the afternoon, with F falling to ~1.6 (both channels significant at $p < 0.002$, Kolmogorov-Smirnov test (KS; Fig. 5B-C).”

Minor points:

1) Line 59 – it is not clear what is meant by “the symbols conveying the message” This terminology is used several more times, so it would be good to define it.

This terminology is a standard part of Shannon’s Information theory and where we use it we reference both the general theory and it’s later applications to Neuroscience. We have amplified our explanation of this concept where it is first used in the Introduction. We hope this gives a better intuitive understanding for the non-specialist:

“Shannon’s information theory has been used to measure the amount of information carried by neurons using spikes, but these are not the symbols that transmit information across the synapse. There the essential symbol is the vesicle and the experimenter needs to observe the release of vesicles conveying the message {Shannon, 1948 #202;Borst, 1999 #203}. This has recently been achieved by multiphoton imaging of the glutamate reporter iGluSnFR{Marvin, 2013 #29} in the retina of larval zebrafish, where it is found that the visual message transmitted from bipolar cells does not use a simple binary code but is instead composed of a number of symbols, formed by one, two, three or more vesicles released as one event{James, 2019 #22}.”

2) Figure 2 – In the results it is stated that both the morning and afternoon time frames are 2 hours, but in the figure legends it says morning is Z1-3 and afternoon is Z6-9. Please correct the discrepancy. Also please clarify here what quanta/cycle means (is this cycle of the sine wave variation in contrast?).

Thanks for pointing out this mistake. We have corrected the legend to make it clear that the 2-hour time windows are ZT0-2 and ZT 6-8. The text is corrected as follows: *“Synaptic function was compared over a two-hour period centred on ZT = 1 hour (“morning”) with a two-hour period centred on ZT = 7 hours (“afternoon”; Fig. 1G).”*

Yes, quanta /cycle refers to the *“total number of quanta released within a single cycle of the sinusoidal stimulus”* and this has been added to the legend of Fig. 2. For instance, using a stimulus modulated at 5 Hz we quantify how many vesicles were released within a time window of 200 ms.

3) Figure 3 – please make a clearer visual distinction between the two control responses in Fig 3 B and C and add them to the figure legend. It is critical to be able to compare the responses before and after agonist/antagonist application.

Thank you for pointing this out. The figure has been improved by superimposing the control responses to the graph. The figure legend has also been corrected/improved to more clearly describe the different conditions.

4) Line 268-269 Can you rephrase this sentence – I’m not sure what it is trying to say: “Notably, the variability of synaptic output was higher than expected for a Poisson process, for which the Fano factor is one.”.

Dating back to the work of Bernard Katz, a central idea has been that vesicle fusion is a process that can be described by Poisson statistics. The fundamental property of a Poisson process is that when it is sampled, the variance in the samples is equal to the mean. In other words, the Fano factor for a Poisson synapse is expected to be one (because Fano factor = variance/mean). The fact that the Fano factor is much greater than one indicates that the ribbon synapse of bipolar cells does *not* adhere to the conventional model of synaptic function in which all vesicles are released independently. We have added this last sentence to the text.

“Through both ON and OFF channels, the variability of synaptic output was significantly higher than expected for a Poisson process, for which the Fano factor is one. In other words, the ribbon synapse of bipolar cells does not adhere to the conventional model of synaptic function in which all vesicles are released independently. “

Reviewer 3

Summary

The paper is extremely well structured and a delight to read. It has three basic parts. 1) Neural response amplitudes in bipolar cells of the inner retina of zebrafish larvae are strongly circadian, with the largest responses occurring in the early afternoon. The reviewer doesn't know that this is elsewhere published, and the finding would strongly influence the way Neurophysiological studies are conducted in the future. 2) Bipolar-cell glutamate release is circadian. There are 4 aspects of this release identified that affect synaptic ability to convey information: i) spontaneous release rate, ii) variation in signal amplitude to fixed stimuli or Fano factor, iii) the temporal precision or ability to phase-lock the stimuli. iv) the types of symbols or tokens released in response to stimulation. 3) All these four factors controlling the efficiency of synaptic information transfer in the inner retina are under control of the diurnal control of the neuromodulator dopamine.

Vesicle release is not Poisson with a Fano factor equal to 1.0 but typically involves Fano factors greater than 1, suggestive of vesicle clustering, and ultimately, multivesicular release. Multivesicular release of up to 5 vesicles simultaneously occurs, and the various packets of vesicles released comprise the multiple symbols of information conveyed by the synapse. The 5-vesicular packet is favored in the afternoon, has the least temporal jitter, and conveys the most information. This finding provides a definite role for multivesicular release. The net impact is that one should not be looking for stable inner retinal circuits in the inner retina to process information. They vary by the hour under neuromodulator control. The net suggestion, though not explicitly stated, is that the bipolar-to-ganglion-cell synapse is the information bottleneck in the retina. The outer retina responses of bipolar cells, which may involve only a few synapses, appear quite stable, although diurnal. While there is a long history of diurnal dopaminergic synaptic modification in the outer retina, this is the first exploration of the effects of dopamine on the inner retina synapses of bipolar cells.

Thank you for your very positive comments. We appreciate your careful reading of the manuscript and help with improving clarity and grammar.

Minor comments:

Abstract:

Neither the abstract nor the title mentions the species.

We now mention the species (larval zebrafish) in the abstract.

Introduction:

-- line 42 'modulators vs modulations'. The words look too similar. Maybe substitute 'changes' for modulations.

Thank you. Now corrected.

--line 43 'Diurnal control of temporal contrast processing' Is the meaning diurnal or circadian?

Apart from Fig. 1, this study concentrates on a comparison of vision in early-morning and early afternoon rather than the whole light-dark cycle so we generally prefer the word "diurnal".

There is a long history of circadian processes modulated by dopamine in the outer retina, which is passed over in the introduction (HJ Wagner, 1980).

This paper is now cited in the Introduction.

Results:

--line 81 'Zeitgeber' The reviewer had to look this word up, perhaps other readers would also. In general it appeared to have a circadian sense (as opposed to diurnal).

Wikipedia defines the term *Zeitgeber* as follows: “A **zeitgeber** is any external or environmental cue that entrains or synchronizes an organism's biological rhythms, usually naturally occurring and serving to entrain to the Earth's 24-hour light/dark and 12-month cycles”. It is therefore appropriate in the context of Fig. 1 because there we map function over several time points during the circadian cycle referencing time to the experimentally imposed light/dark cycle. We have tried to clarify the relationship between the timing of the light/dark cycle and the term *Zeitgeber* as follows: “When animals were placed on a cycle of 14 hours light and 10 hours dark, no significant synaptic responses could be detected in the 6 hours before light onset (at *Zeitgeber* times 18-0 hours), consistent with previous observations that larvae are blind at night{Emran, 2010 #214}.”

--line 116 'contrast' The reviewer identifies this term with temporal sine waves over a mean luminance. These are step functions on no background.

Thanks for highlighting this lack of clarity. We did indeed use sinusoidal stimuli for measurements of both presynaptic Ca and glutamate release but did not clearly indicate this in the stimulus trace for the Ca measurement in Fig. 1E. This has now been altered to show the modulation in intensity around the mean rather than the contrast calculated from this modulation.

-line 120 'C1/2 it fell' Fix grammar.

Thank you. Grammar has been corrected.

--line 134 'individual active zones vesicles' unclear 'vesicles from individual active zones'?

Thank you. This sentence has been corrected

--line 158 'at an individual OFF active zone' 'an individual OFF-bipolar active zone'

Thank you. This sentence has been corrected.

--line 201 'The dynamic range over which D1 receptors adjusted synaptic gain' is this synaptic gain or synaptic output. The changes seen might originate in distal retina, as seen in Fig. 1.

We have now defined the way we use the term “synaptic gain” more explicitly at it's first mention in the Results section, as follows: “There was little diurnal modulation of the CRF measured at ON synapses but in the OFF channel the synaptic gain, measured as the maximum rate of vesicle release at 100% contrast, increased from 15.25 ± 2.5 vesicles/s in the morning to 25.5 ± 1.5 vesicles/s in the afternoon (Fig. 2C and D).”

We of course completely agree that these changes might reflect effects at a number of sites in the retinal network and this was discussed in part of the Discussion called “Diurnal modulation of gain’ which has now been expanded.

--line270-1 'The spike responses of post-synaptic RGCs are less variable, with a Fano factor as low as 0.3 at higher contrasts30' But Berry shows a Fano factor of 1.5 at low contrasts.

We have now given a more detailed description of the results of Berry as follows: “The spike responses of post-synaptic RGCs are generally less variable, with a Fano factors varying from 1.5 at low contrasts down to 0.3 at higher contrasts{Berry, 1997 #61}”.

Line 441 '...an increase in vesicle release rate is associated with an increase...' The 'i' in 'increase' number two needs italic font.

Thank you. Corrected

Methods

Line 655 'E2 medium' Specify source or composition.

The composition of the E2 medium is now specified in the first subsection of Methods.

Line 671 'Full-field light stimuli were generated by an amber LED ($I_{max} = 590 \text{ nm}$ Thorlabs), filtered through a 590/10 nm BP filter (Thorlabs),' Then most of what is described is lws cone physiology. This maybe should be discussed. There are no UV cone signals. The experiments don't determine whether the UV pathway is dopamine sensitive, for instance. As noted 'These wavelengths will most effectively stimulate red and green cones.'

Yes, the wavelengths stimulate red and green cones (roughly equally) but do not stimulate UV cones which in zebrafish UV are enriched in the strike one which underlies prey capture {Yoshimatsu, 2020 #256}. This is now noted in Methods. It would certainly be interesting to carry out a similar study that investigates diurnal regulation of prey-capture behaviour and the potential retinal mechanisms.

--line 688 '...200 nM (Sigma).Finally,...' typo.

Thank you. Corrected.

--line 699 '...where t_{qi} is the time of the i th q -quantal event,...' How is t_{qi} determined?

At this point in Methods we have now inserted the sentence "The method for estimating t and q is summarized in Supplementary Figure 4." This figure (previously Fig. S3) describes the basic steps in the analysis sequence for the quantal decomposition of iGluSnFR signals (which are explained and validated in much greater detail in James et al. 2019). The legend to Supplementary Figure 4 has now been improved to tie-in better with the reference to t_{qi} in Methods.

Supplementary Figure 4

Figure S4. Decomposition of iGluSnFR signals into vesicle counts

Summary of the basic steps for quantal decomposition of iGluSnFR signals. Each event is assigned a time, t , and an estimate number of quanta, q , using the following basic steps.

1. Raw trace extracted from individual active zones (linescan, 1 KHz).
2. Trace deconvolved using the estimated Wiener filter and thresholded to extract events above noise. The timing of the peak is the event time t . The amplitude of the event is the peak of the deconvolved trace at time t .
3. The distribution of event amplitudes is plotted and fitted by the sum of Gaussians with peaks differing by integer multiples of a quantal value q . This example is a histogram of event amplitudes for an active zone in which 373 events were accumulated using stimulus contrasts of 20%, 60% and 100% and a frequency of 5 Hz. The black line is a fit of eight Gaussians, identified using a Gaussian mixture model. Note that the variance of successive Gaussians did not increase in proportion to the peak number. The first peak had a value of 0.24, and the distance between peaks averaged 0.25, indicating the existence of a quantal event equivalent to ~ 0.25 .
4. Maximum-likelihood estimation of the number of quanta, q , in each event based on its amplitude in 2 and the distribution in 3.

These steps are explained and validated in much greater detail in James et al. (2019).

--line 700 'number of events of composed of q-quanta.' Typo

Thank you. Corrected

Fig.1 legend

A. 'light step' Although the 590nm wavelength is buried in the methods, which this Journal leaves until last, it would be helpful to include the wavelength in the legend, and mention that it is the lws cone pathway that is being studied.

This is now specified in the legend

B. --line 95-6 '...region of showing terminals...' typo

Thank you. Corrected.

E. define 'contrast' as used in used in this figure. The stimuli are all step functions, presumed 100% contrast.

These were not step functions but sinusoidal modulations around a mean. The stimulus trace we provided was in units of contrast, which was misleading. The stimulus trace has now been adjusted to represent intensity as a function of time the point that this is a sinusoidally modulated stimulus has also been emphasized in the legend.

F. --line 109 'h=7.0' Really?

Yes!

--line 113 'All error bars show ± 1 SD.' Error bars are SD (not SE?)

Thanks for spotting this mistake. The error bars indeed 1 SE with exception of fig 1D and this has been corrected.

The legend should mention the larval age.

Added "Zebrafish larvae were 7-9 days post-fertilization."

Figure 2 legend

A. In this and succeeding figures, it isn't clear how ON and OFF responses are distinguished using sinusoidal stimuli, or is this determined in separate experiments with other stimuli?

Thank you for pointing out this – we certainly should have made this clearer. Where we introduce the use of iGluSnFR we now state: "To measure transmission of the visual signal in terms of its elementary units - synaptic vesicles - we expressed the reporter iGluSnFR{Marvin, 2013 #29} sparsely in bipolar cells (Fig. 2A). A variety of morphological types were investigated in this study but we focused on a comparison on the two most basic functional types, ON and OFF cells, identified through their responses to steps of light (Supplementary Figure 3). Fig. S3C then shows how steps of light were applied immediately before the 5 Hz stimulus to allow easy identification of ON and OFF cells, as show below.

Figure S3. Types of bipolar cell sampled in this study

A. Multiphoton section through the eye of a larval zebrafish (7 dpf) expressing iGluSnFr in a subset of bipolar cells. **B.** Examples of some of the different morphological subtypes of OFF (top) and ON (bottom) bipolar cell which were sampled in this study. **C.** We focused on a comparison on the two most basic functional types, ON and OFF cells, identified through their responses to steps of light applied before applying stimuli modulated at 5 Hz. iGluSnFR signal were measured in a total of 91 synapses in ON and 151 synapses in OFF cells.

--line 141-2 '...using a stimulus of variable contrast..' using stimuli of variable contrast

Thank you. Corrected.

--line 158 'at an individual OFF active zone' 'an individual OFF-bipolar active zone'

Thank you. Corrected.

Fig. 3 legend

Specify the concentration of SCH23390 in the legend.

Thank you. Now the final concentration of SCH 23390 in the eye (20 nM) has been included in the legend.

--line 190 'Each point shows the mean \pm s.e.m.' Here you give 'sem' is that also true of Fig. 1, where 'SD' was given? The methods say sem unless otherwise noted.

The mistake was in legend of Fig.1 which has been corrected. As stated in Methods, all errors were \pm 1 sem.

--line 193 C. 'synapses in three conditions: afternoon (green dots),' There are no green dots in panel 'C.'

Thank you. The figure and legend have been corrected.

Fig. 4 legend

In B. and C. it appears to the reviewer that event rate should be the X-axis and the Q_e should be the Y-axis. It is hard to contemplate that the Q_e determines the spontaneous event rate.

There is no independent variable in these plots. They are essentially distributions that could have been plotted as bar charts as in Fig. 7B-E (except that the comparison of three conditions prevented us from using bars because it makes visualization very difficult). We have therefore left as is to maintained consistency with other figures.

Fig. 6 legend

--line 307 '...afternoon (green dots; n = 14 synapses) during the (open green dots; n= 10 synapses) and...' Add 'morning'

Thank you. Legend corrected.

Fig. 7 legend

B, C, D, E, The symbol legends indicate that two items with different color codes are being compared, but there are 3 colors on the bar graphs: gray and two shades of either red or green. The reviewer couldn't read the bar charts.

Thank you for spotting this mistake. Now we modified the figure (see below). Light grey bars are now used for the "Morning" condition and dark grey bars for the "afternoon + SCH 23390" condition.

Suppl Fig S2

Panel B. Four traces but only two colors. The trace labels could be put over individual traces rather than all in a single box.

Thank you for pointing this out. The figure has been improved according to the reviewers suggestions.

Reviewer 4

The manuscript by Moya-Diaz and colleagues describes a most interesting case of neuromodulator mediated neural network adjustment in the retina. The authors convincingly show that synapses of bipolar cells shown diurnally regulated visual processing, affecting synaptic gain, variability, and noise. The efficiency of synaptic transmission is modified by potentiation of multivesicular release. These new results are satisfyingly linked to the large literature on the role of the neuromodulator dopamine in retinal plasticity. The results are well presented and the conclusion are both supported by the data and novel. Overall this is a fantastic paper using an original experimental approach to settle an interesting question. However, before the study can be published there are many points that need to be addressed, many of them minor, but also a couple major ones as detailed in the following:

We are very pleased that you found this work interesting.

Major points:

1. There are major problems with Figure 1:

In Figure 1 BCD, as it is indicated in the legend, the data from B is plotted in C. However there must be a mix-up, including a wrong color label. Although the y axis is the same, the red line in C is probably okay, however the blue line in C should be black. It is not clear what data is plotted by the black line in panel C and certainly absent from panel B.

Thank you for pointing out the mix-up with the colour coding of the lines. We pasted the wrong version of B into Figure 1. The correct version is below. These traces match with the irradiance/response function presented in C.

Information in panel D plots data from panel C, but here the sensitivity is highest at ZT16, which is not possible according to B and according to the fact that they larvae are more or less blind at ZT16.

We cannot see an incompatibility between C, D and E of Fig. 1. The data in B and C show the measurement of luminance sensitivity for three time points: 1, 6 and 10 h. The data in D extend the time-course to include measurements at 13 and 16 h. The larvae are not blind in the first hour or so after the lights turn off. Fig. 1 does not show measurements of luminance sensitivity at 19 h and 22 h because it is not possible to measure because terminals stop generating responses. This was shown in Fig. S1A, where we plot the percentage of terminals

that generate a significant response at different times during the light-dark cycle. At 16 h, 15% of terminals responded significantly to the brighter light steps allowing us to make a luminance-response plot as in Fig. 1C. But at 19 h and 22 h no terminals generated significant responses.

Also in D, I am really wondering if the y axis is scaled as in the figure (0.1, 1,10,100), what scale will be taken for the error bars? The reason I ask is that the error bars looked extraordinarily big in this figure. IS SD or SEM? Is the logarithmic scale an issue? Hence Figure 1 must be reworked and corrected.

The error bars are 1 SD but a mistake was made in recalculating the SD after taking the inverse of the original measurements of $1/I_{1/2}$. The corrected graph is shown below.

2. Figure 2 is at best misleading:

I am confused on how the data depicted in panel E can be squared with the data depicted in B and C. The authors should check their plot, and if correct explain panel E better. (For instance, I am confused how the similar response at 20% contrast in panel C relates to the very different values in panel E. It may be my lack of understanding, but then a clearer explanation is warranted.)

Thanks for highlighting this lack of clarity. The data in C and D are derived from the measurements in B (i.e glutamate release at different contrasts). The key transformation from C and D to E is differentiation, yielding a relation called “contrast gain”. So the large difference between OFF morning and OFF afternoon around 20% contrast in E reflects the very different slopes of the contrast-response functions around 20% contrast in C.

The reason for re-expressing the contrast response functions in C and D as “contrast gain” in E is that it gives insight to a key property of any sensory system – it’s ability to discriminate one stimulus from another. The plots in E therefore make what we believe are two very important points. First, that in the morning the ON pathway discriminates changes in contrast better than the OFF pathway, at least over the range most common in nature (<40%). Second, the OFF pathway experiences much stronger diurnal modulation than the ON. Third, this

causes the OFF pathway to discriminate changes in contrast better than the ON in the afternoon, especially at contrasts below 40%.

We have provided a little more explanation as well as two key references {Dayan, 2005 #258; Smith, 2009 #257}. This section now reads:

“This increase in synaptic gain was accompanied by an increase in contrast sensitivity, and the combined effects were assessed as the derivative of the CRF, which we term “contrast gain” (Fig. 2E). The reason for re-expressing the contrast response functions as “contrast gain” is that this gives insight into a key property of the visual system – its ability to discriminate one stimulus from another{Dayan, 2005 #258; Smith, 2009 #257}. Contrasts in natural visual scenes rarely exceed 40%{Zimmermann, 2018 #198} and in the morning changes in this range was signalled best through the ON channel. But in the afternoon the OFF channel became dominant, with contrast gains increasing by factors of 2-6. These results indicate that diurnal modulation of retinal processing adjusts the relative importance of ON and OFF pathways in signalling temporal contrast.”

Minor points:

Line 43-44: “focusing on the visual signal transmitted that bipolar cells transmit to the inner retina.” Delete “transmitted”

Done.

Line 63 “instead composed of a number of symbols, composed of one, two, three or more” replace one “composed”

Done

Line 82, “larvae are blind at subjective night” The term “subjective” is used by conventional chronobiologists exclusively when referring to time without any zeitgebers. Delete “subjective”.

Done

Line 86.87 “Over the course of the day, luminance-sensitivity increased gradually over a range greater than 200-fold (Fig. 1D).” this may need to be edited because of the points mentioned related to the figure BCD.

This has been checked and is still correct, so has been left as is.

Line 95, ensure abbreviations like “IPL” are at least spelled out once at first appearance in the text.

The term “IPL” is now defined in the legend of Fig 1. Other abbreviations have also been defined on their first appearance in the text.

Line 103, “red arrow” should be “dashed red arrow”.

Thank you. Corrected.

Line 106, the sample size should be indicated and the ON or OFF terminals should be specified.

This information has been added to the figure legend.

Line 134-136 “Synaptic function was compared over a two-hour period beginning 1 hour after light onset (“morning”) with a two-hour period beginning 6 hours later (“afternoon”); Fig. 1G). This reads slightly confusing. It is better to specify the recording period in Zeitgeber time, like ZT1-ZT3, ZT7-ZT9 .

We clarified this point as follows: “Synaptic function was compared over a two-hour period centred on ZT = 1 hour (“morning”) with a two-hour period centred on ZT = 7 hours (“afternoon”); Fig. 1G).”

Line 143 “ZT 6-9 hours”, probably ZT7-ZT9 is meant – please verify.

Thank you. The right ZT window is “ZT 6-8 hours”. Now corrected (see above)

Line 145 The legend of figure 2C should contain some information on the statistical treatment

This has been added.

Line 155, what do the error bars represent in panels C and D? Statistical information missing?

Error bars represent S.E.M. Now included in the legend as well as statistical information.

Line 168-170, “Contrasts in natural visual scenes rarely exceed 40% and in the morning this range was signalled best through the ON channel. But in the afternoon the OFF channel became dominant, with contrast gains increasing by factors of 2-6.” This conclusion may need to be clarify a bit better, including the value of 2-6 needs to be more clearly specified.

We have now highlighted the range up to 40% contrast with a grey box and amplified our presentation of these results as follows:

“The increase in synaptic gain was accompanied by an increase in contrast sensitivity, and the combined effects were assessed as the derivative of the CRF, which we term “contrast

gain" (Fig. 2E). The reason for re-expressing the contrast response functions as "contrast gain" is that this gives insight into a key property of the visual system – its ability to discriminate one stimulus from another{Dayan, 2005 #258; Smith, 2009 #257}. Contrasts in natural visual scenes rarely exceed 40%{Zimmermann, 2018 #198} and in the morning changes in this range were signalled best through the ON channel (grey box in Fig. 2E). But in the afternoon the OFF channel became dominant, with contrast gains through this channel increasing by factors of 2-6 at contrasts up to 40%. These results indicate that diurnal modulation of retinal processing adjusts the relative importance of ON and OFF pathways in signalling temporal contrast."

Line 193, "in ON bipolar cell synapses in three conditions: afternoon (green dots)" - there are no green dots. "afternoon (green dots)" should be deleted. "three" to "two"

Thank you. Figure legend has been corrected.

Line 204-205 , "But diurnal modulation of gain was narrower than this potential range: 1.7-fold in OFF synapses and 1.1-fold in ON." But there is no diurnal modulation at ON response according to line 163" There was little diurnal modulation of the CRF measured at ON synapses" and the statistics information in the legend of Figure 2 D. But if read line 204-205 alone, it sounds there is a significantly difference between morning and evening in ON response, but the difference is very small (the ratio is only 1.1).

The reviewer is absolutely right. The factor of 1.1 in Fig. 3D was obtained from the ratio of the fits to the contrast-response functions and does not represent a statistically significant difference in the measured data. This sentence has therefore been simplified to "But diurnal modulation of gain was narrower than this potential range with a maximum of 1.7-fold in OFF synapses.". The arrow indicating the asymptotic diurnal change in gain in ON synapses estimated from the fits has also been removed from Fig. 3D to avoid confusion.

Line 245, what do the error bars mean in B and C? Any statistics information?

Error bars indicate S.E.M. Statistical analyses were performed using "One-way ANOVA" test for comparing 3 groups. Statistical differences were observed in OFF synapses ($p < 0.0001$) but not in ON ($p < 0.37$). Now added to the figure legend.

Line 266, KS test stands for Kolmogorov-Smirnov test.

Yes. Now defined in the text and legend.

Line 266-267 , "The increase in contrast gain and sensitivity in the afternoon (Fig. 2C-D)" in D there is no increase in the afternoon for ON response.

Thanks for encouraging precision. We have reworded this part as follows: "In the OFF channel, the increase in contrast gain and sensitivity in the afternoon (Fig. 2C) was therefore

also associated with increased reliability of bipolar cell synapses. Synapses in the ON channel did not undergo significant changes in gain and sensitivity (Fig. 2D) but also became more reliable.”.

Figure 5: Panels D and E do not add much to panels B and C. It looks more convincing, but simple because they ignore the high variability in B and C by averaging the values again.

That is correct. We have elected to leave the figure unchanged because B and C show something that D and E do not: a tendency for the Fano factor in the afternoon to fall at higher contrasts. Berry et al. (1997) demonstrated the same tendency when measuring the variability of spike responses in post-synaptic ganglion cells (a point remarked on by Reviewer 3). We do not highlight this aspect of the results in the text but hope that aficiannado’s will be able to make the comparison.

Line 281 Specify the meaning of the error bars in panels B and C.

Error bars indicate S.E.M. Now added to the legend.

Line 287-288 t-test should not be used for 3 groups.

We reanalysed statistics comparisons for 3 groups by using One-Way ANOVA tests ($p < 0.0001$, in ON and OFF synapses). Now included in the legend.

Line 290 Figure 6: The temporal precision of MVR is under diurnal control in the OFF channel” But this figure is not only about MVR.

We have reworded to: “The temporal precision of vesicle release is under diurnal control in the OFF channel”.

Line 297, legend for B: what does the dash line mean?

The dash line is showing the temporal jitter value (ms) of events composed by 5 or more quanta. Now clarified in the legend.

Line 306-310, legend for C: what does the dash line mean?

The dashed line shows the temporal jitter value (ms) of events composed of 8 or more quanta. Now clarified in the legend.

“The relationship observed in the morning is better described by a straight line with $a = 34.7 \pm 1.5$ and a slope = -3.6 ± 0.5 .” there is no fitting in the figure for morning data, but afternoon data and morning +D1 agonist data.

This fit has now been added to the figure (dashed green line), and fitting parameters for each condition added to the legend.

Line 322, (Fig. 3H; t-test at each Q_e). I guess it meant Figure 6 C and t test should not be use with 3 groups in the figure.

Thank you for spotting this mistake. Indeed, this should have been Fig 6C. However, in this case the t-test is properly used as the word “diurnal” is referring to the comparison between morning and afternoon conditions.

“Diurnal modulation of temporal precision was weaker in ON synapses and only significant for events composed of 1-3 vesicles (Fig. 6C; t-test at each Q_e)”

Line 325-326 “Diurnal variations in dopamine therefore modulate the temporal accuracy of vesicle release” should add “for OFF synapses”

Corrected.

Line 338 Legend for A, is this an OFF or an ON terminal?

It is an OFF terminal. Now added to the legend.

Line 340-347 legend B-E n should be specified.

Thank you. Numbers are specified now in the legend.

Line 342 legend for Figure 7B “($p < 0.059$, Chi-squared test)” and line 352 “(Fig. 7B; $p < 0.05$, KS test)” does not fit.

Thanks for spotting this mistake. The correct value is $p < 0.05$ (KS-test).

Line 346 legend for D “ $p < 0.007$ ” and line 354 “(Fig. 7D; $p < 0.02$)” does not fit.

Thanks for spotting this mistake. The correct value is $p < 0.02$ (KS-test).

Line 359-361 *“Qualitatively similar modulation of MVR was observed over a range of contrasts from 20% to 80% and blocking D1 receptors in the afternoon shifted the distribution back to univesicular release in both ON and OFF channels (Fig. 7D and E).” The data is not shown for other contrasts, so it is needed to be indicated as “data not shown”. Furthermore, the second half of the sentence sounds for me that the effect of blocking D1 receptors at other contrast levels have also been checked. If this is the case, “data not shown” can be added here, or if not, “at 60% contrast” should be added.*

Thank you for pointing out this. We have added “data not shown” to this sentence.

Line 361-362, *“Diurnal variations in dopamine therefore modulate MVR.” I think should be “Diurnal variations in dopamine therefore modulate the probability or distribution of MVR”*

Thank you. Corrected

Line 365 *“How do changes in synaptic gain (Figs. 2-3), noise (Fig. 4-6) and MVR (Fig. 7)” noise is only investigated in figure 4.*

We have changed the word “noise” to “variability”, different aspects of which are studied in Figs. 4-6.

Line 392 *“(270%; $p < 0.001$; Fig. 8B)” which test?*

KS test. Added.

Line 394 *“release rate of 2.5 vesicles s⁻¹ around C1/2,” Where does this value come from?*

Fig. 2C. Added.

Line 415 legend for Figure B, *what do the error bars represent? Sample size? Any statistics information?*

Error bars represent S.E.M. Number of synapses per condition and statistical analyses (One-way ANOVA) have been now included in the legend.

Line 417-419 legend for C and D, *for each figure, there is only one sample size provided. Does it apply to both recordings in the afternoon and morning? Is the curve fitted according to morning values or afternoon values? What do the error bars signify?*

Thank you for spotting this. The number of synapses per condition has been added to the legend now (Afternoon, OFF=33: ON = 13; Morning, OFF: 15, ON:10). The fitting is according to afternoon values. Now clarified in the legend. Error bars represent S.E.M.

Line 472-474 *“Most strikingly, reduced variability of synaptic responses and increased emphasis on MVR increased information transfer through the ON pathway without an increase in synaptic gain.” This reads strange and should be reworded.*

We have reworded this sentence and attempted to link it better with the previous as follows:
“Dopamine plays a major role in regulating all these aspects of retinal function although the relative contributions of these mechanisms differed between ON and OFF pathways. In

the ON pathway, for instance, the reduced variability of synaptic responses and increased emphasis on MVR increased information transfer without an increase in synaptic gain.”

Some additional recent references on the effect of circadian changes on vision in the discussion would also improve the manuscript a bit.

We have added this sentence at the start of the Discussion which points the reader to four recent reviews on this topic: *“The physiology of this circuit is regulated by circadian clocks intrinsic to photoreceptors and neurons in the inner retina and these become entrained by light and dopamine{Storch, 2007 #259; Besharse, 2016 #64; Li, 2019 #156; Goel, 2021 #261}.”*

Besides these many small issue, this is an excellent study.

Thanks for your suggestions and helping us improve its presentation!

REVIEWERS' COMMENTS

Reviewer #1 (Remarks to the Author):

The authors have addressed my comments sufficiently. I have no further comments.

Reviewer #2 (Remarks to the Author):

The authors have addressed my concerns.

Reviewer #3 (Remarks to the Author):

On a second reading this is a very powerful paper, a dense read, and well worth the effort.

Minor points:

79. 'we began by imaging synaptic activity in bipolar cells with SyGCaMP226 (Fig. 1A).' For consistency with the image in Fig. 1A 'synaptic activity in bipolar cell synaptic terminals'. (The image shows the inner plexiform layer, not BC cell bodies.)

128. 'At ZT = 16 hrs:' typo: 'At ZT = 6 hrs:'

Fig. 1F. A dashed blue arrow pointing to half maximal contrast for ZT 13 h is missing.

204. 'One-way ANCOVA-test, $p < 0.53$ ' typo. ' $p = 0.53$ '.

Fig. 2E. The figure panel lacks a legend (on the image) explaining the treatments in the different line types.

267. 'Relative response gain by diurnal modulation' Not sure what this means. Is it a 'Diurnal ratio'.

326. 'Each point shows the mean \pm s.e.m. (One-Way ANOVA, $p < 0.0001$)' Ok, this has to be a 2-way ANOVA because one is testing both for the reduction in frequency with increased event amplitude and for the difference between ADTN treatment and control. There is also question about the use of line segments connecting points in Figs. 4B and 4C. Since the X-axis is quantal, there are no values to be interpolated between points. Really, this is a histogram plot. Logically there would be a t-test and p-value for each event amplitude, or a 1-way ANOVA if all three treatments are considered.

349 'In the morning, F was ~ 2.6 in both ON and OFF synapses when averaged over a range of contrasts, but synapses were more reliable in the afternoon with F falling to ~ 1.6 (both channels significant at $p < 0.002$, Kolmogorov-Smirnov test (KS); Fig. 5B-C).' Kolmogorov-Smirnov is not strictly a test for a difference in the means, but a combination of means, variances, and distribution shape. It's a fine point but maybe one should say the distributions are significantly different.

441. 'Increasing activation of D1 receptors in the morning reduced temporal jitter in events composed for multiple quanta in OFF synapses' typo: 'composed of multiple quanta'.

491. 'In the morning, MVR was more prevalent in OFF synapses' Clarification: 'more prevalent in OFF synapses than ON synapses'

567. 'Mutual information I (S;Q) in four conditions' This is 'I(S:Q)' as in Fig. 8B Y-axis title.

567-570 (i), (ii), (iii), (iv) aren't used in the Fig. 8C panel. Use Morn (AM), After (PM) etc as on the Fig. 8C X-axis.

808. 'islet2b::mGCaMP6f' Extra ':' needed?

832. In Fig. s3A, are these projections of image stacks?

836. 'iGluSnFR signal were measured in a total of 91 synapses' typo: 'signals'

864. 'Each event is assigned a time, t, and an estimate number of quanta, q,' typo: 'estimated'

867. 'deconvolved' Which function is extracted from the raw data? Glutamate release rate?

897. 'Tg(ribeye::Zf-SyGCaMP2)' is ';' needed?

Reviewer #4 (Remarks to the Author):

The authors have resolved all the issues I raised. Congratulations to a really interesting paper.

Response to Reviewer 2

Thank you for your helpful suggestions.

79. *'we began by imaging synaptic activity in bipolar cells with SyGCaMP226 (Fig. 1A).'* For consistency with the image in Fig. 1A *'synaptic activity in bipolar cell synaptic terminals'*. (The image shows the inner plexiform layer, not BC cell bodies.)

Thank you. Now the text has been corrected.

128. *'At ZT = 16 hrs:'* typo: *'At ZT = 6 hrs:'*

Thank you. Corrected.

Fig. 1F. A dashed blue arrow pointing to half maximal contrast for ZT 13 h is missing.

Thank you. Corrected as below.

204. *'One-way ANCOVA-test, $p < 0.53$ ')* typo. *' $p = 0.53$ '*.

Thank you. Corrected.

Fig. 2E. The figure panel lacks a legend (on the image) explaining the treatments in the different line types.

Thank you. Corrected as below.

267. 'Relative response gain by diurnal modulation' Not sure what this means. Is it a 'Diurnal ratio'.

Yes, the relative response gain is now defined more clearly in the text, as follows:

"The dynamic range over which D1 receptors adjusted synaptic gain was calculated as the ratio of the CRFs in the presence of the agonist and antagonist ("relative response gain"): in both ON and OFF channels the maximum modulation was ~16-fold, occurring at contrasts of 20-40% (Fig. 3D)."

Thanks for pointing out this mistake - we did indeed run a two-way ANOVA. However, we

There is also question about the use of line segments connecting points in Figs. 4B and 4C. Since the X-axis is quantal, there are no values to be interpolated between points. Really, this is a histogram plot.

Thank you. The line segments connecting the points have been removed as below.

326. 'Each point shows the mean \pm s.e.m. (One-Way ANOVA, $p < 0.0001$)' Ok, this has to be a 2-way ANOVA because one is testing both for the reduction in frequency with increased event amplitude and for the difference between ADTN treatment and control.

Logically there would be a t-test and p-value for each event amplitude, or a 1-way ANOVA if all three treatments are considered.

We have gone for a simpler presentation in which we provide the statistics for a single metric in each condition – the total rate of spontaneous vesicle release (i.e integrating vesicles across events of all amplitudes). Differences between conditions are then summarized as mean \pm sem of this metric and conditions compared with a one-way ANOVA. The statistical tests are now described in the text (not the figure legend) as follows:

"Integrating across events of all amplitudes, the average rate of spontaneous release in OFF synapses was 1.95 ± 0.04 vesicles s^{-1} in the morning, falling to 0.37 ± 0.02 vesicles s^{-1} in the afternoon (Fig. 4B; difference significant at $p < 0.0001$, one-way

ANOVA). In ON synapses these values were 1.47 ± 0.09 vesicles s^{-1} and 0.46 ± 0.02 vesicles s^{-1} ($p < 0.0001$, Fig. 4C). Across both channels, therefore, spontaneous noise was 3-5 times lower in the afternoon compared to the morning.

Spontaneous release in OFF synapses was modulated by dopamine. Increased activation of D1 receptors by injection of ADTN suppressed spontaneous release in the morning to levels close to those measured in the afternoon (0.51 ± 0.03 vesicles s^{-1}), a change significant at $p < 0.0001$ (one-way ANOVA; Fig. 4B). In contrast, ON ADTN had no significant effect on spontaneous release from ON synapses (1.66 ± 0.18 vesicles s^{-1} , $p = 0.3$; Fig. 4C)."

349 *'In the morning, F was ~2.6 in both ON and OFF synapses when averaged over a range of contrasts, but synapses were more reliable in the afternoon with F falling to ~1.6 (both channels significant at $p < 0.002$, Kolmogorov-Smirnov test (KS); Fig. 5B-C).' Kolmogorov-Smirnov is not strictly a test for a difference in the means, but a combination of means, variances, and distribution shape. It's a fine point but maybe one should say the distributions are significantly different.*

Thanks for pointing this out. We have removed reference to the KS test.

441. *'Increasing activation of D1 receptors in the morning reduced temporal jitter in events composed for multiple quanta in OFF synapses' typo: 'composed of multiple quanta'.*

Thank you. Corrected.

491. *'In the morning, MVR was more prevalent in OFF synapses' Clarification: 'more prevalent in OFF synapses than ON synapses'*

Thank you. Corrected

567. *'Mutual information I (S;Q) in four conditions' This is 'I(S:Q)' as in Fig. 8B Y-axis title.*

Thank you. Corrected.

567-570 (i), (ii), (iii), (iv) *aren't used in the Fig. 8C panel. Use Morn (AM), After (PM) etc as on the Fig. 8C X-axis.*

Thank you. Corrected

808. *'islet2b::mGCaMP6f' Extra ':' needed?*

Thank you. Corrected.

832. *In Fig. s3A, are these projections of image stacks?*

Yes, these pictures are an average of 5 images from the same plane.

836. 'iGluSnFR signal were measured in a total of 91 synapses' typo: 'signals'

Thank you. Corrected.

864. 'Each event is assigned a time, t, and an estimate number of quanta, q,' typo: 'estimated'

Thank you. Corrected.

867. 'deconvolved' Which function is extracted from the raw data? Glutamate release rate?

We have amended this sentence as follows: "*Trace deconvolved using the estimated Wiener filter and threshold crossings used to detect events above noise.*"

897. 'Tg(ribeye:Zf-SyGCaMP2)' is ';' needed?

Thank you. Corrected.